# Theoretical Convergence Analysis for Hilbert Space MCMC with Score-based Priors for Nonlinear Bayesian Inverse Problems

## Abstract

In recent years, several works have explored the use of score-based generative models as expressive priors in Markov chain Monte Carlo (MCMC) algorithms for provable posterior sampling, even in the challenging case of nonlinear Bayesian inverse problems. However, these approaches have been mostly limited to finite-dimensional approximations, while the original problems are typically defined in function spaces of infinite dimension. It is well known that algorithms designed for finite-dimensional settings can encounter theoretical and practical issues when applied to infinite-dimensional objects, such as an inconsistent behavior across different discretizations. In this work, we address this limitation by leveraging the recently developed framework for score-based generative models in Hilbert spaces to learn an infinite-dimensional score, which we use as a prior in a function-space Langevin-type MCMC algorithm, providing theoretical guarantees for convergence in the context of nonlinear Bayesian inverse problems. Crucially, we prove that controlling the approximation error of the score is not only essential for ensuring convergence but also that modifying the standard score-based Langevin MCMC through the selection of an appropriate preconditioner is necessary. Our analysis shows how the control over the score approximation error influences the design of the preconditioner—an aspect unique to the infinite-dimensional setting.

## 1 Introduction

Solving inverse problems is a central challenge in many applications. The objective is to estimate unknown parameters using noisy observations or measurements (Tarantola, 2005). One of the main challenges of inverse problems is that they are often ill-posed (Hadamard, 1923). However, by framing an inverse problem within a probabilistic framework known as Bayesian inference, one can characterize all possible solutions (Tarantola, 2005; Lehtinen et al., 1989; Stuart, 2014). In the Bayesian approach, we first define a prior probability distribution that describes our knowledge on the unknown before any measurements are taken, along with a model for the observational noise. The objective is to estimate the *posterior distribution*, which characterizes the distribution of the unknown given noisy measurements. One can then sample from the posterior to extract statistical information for uncertainty quantification (Stuart, 2014; 2010; Knapik et al., 2011; Dashti and Stuart, 2011).

*Score-based generative models* (SGMs) offer a powerful way to compute the posterior. SGMs are deep learning tools that sample from a complex high-dimensional distribution by first learning the *(Stein) score* (Liu et al., 2016)—the gradient of the logarithm of the probability density function of the distribution—and then using it in various sampling algorithms. A popular version among them, known as *score-matching with Langevin dynamics* (SMLD), uses the learned score in a Monte Carlo Markov Chain (MCMC) algorithm based on Langevin dynamics (Song and Ermon, 2019). Well-documented practical issues with Langevin MCMC samplers, such as slow mixing and inaccurate score estimation in low data density regions, are handled using heuristics inspired by simulated annealing (Kirkpatrick et al., 1983) and annealed importance sampling (Neal, 2001), where data are perturbed with different noise levels, a single score network is trained to estimate scores for all these levels, and during sampling, scores for large noise levels are used initially while the noise is gradually reduced. To enable new sampling procedures, Song et al. (2020) found that SMLD and diffusion-based methods (Sohl-Dickstein et al., 2015; Ho et al., 2020) can be related through a unified framework, based on

stochastic differential equations (SDEs), often referred to as *score-based diffusion models*. Here, instead of perturbing data with a finite number of noise distributions at discrete times, Song et al. (2020) have considered a continuum of distributions that evolve in time according to a diffusion process whose dynamics is described by an SDE. Crucially, the reverse process is also a diffusion (Anderson, 1982) satisfying a reverse-time SDE whose drift depends on the score, which can be estimated through a neural network via score matching (Vincent, 2011; Song and Ermon, 2020).

After their introduction, SGMs have been utilized for solving inverse problems in a Bayesian fashion. Some have proposed to sample from the posterior using the score conditioned on observations (Batzolis et al., 2021; Kawar et al., 2021; Jalal et al., 2021). Others have suggested utilizing the learned score of the prior distribution, that is, the so-called *unconditional score model*. Currently, there are two ways the learned score of the prior is used to sample the posterior distribution of an inverse problem: (i) modifying the unconditional reverse diffusion process of a pretrained SGM, which initially produces samples from the prior distribution, by conditioning on the observed data so that the modified reverse process yields samples from the posterior (Song et al., 2021; You and Dragotti, 2024; Chung et al., 2022); and (ii) using a score-based MCMC sampler, as in the seminal work of Welling and Teh (2011) on stochastic gradient Langevin dynamics, where the score of the prior distribution is learned to capture more complex features (Feng et al., 2023; Sun et al., 2024; Xu and Chi, 2024), thereby extending *de facto* the SMLD algorithm to posterior sampling.

Regardless of whether they utilize the conditional or unconditional score, all the works cited above have something in common: they assume that the posterior is *supported on a finite-dimensional space*. However, in many inverse problems, especially those governed by partial differential equations (PDEs), the unknown parameters to be estimated are *functions* that exist in a suitable function space, typically an infinite-dimensional Hilbert space. Unfortunately, discretizing the input and output functions into finite-dimensional vectors and utilizing standard SGMs to sample from the posterior is not always desirable—it is well known that algorithms designed for finite-dimensional settings can encounter theoretical and practical issues when applied to infinite-dimensional objects, such as an inconsistent behavior across different discretizations (Stuart, 2010). In the last year, however, some progress has been made to address these concerns. Building upon the theory of infinite-dimensional stochastic analysis (Föllmer and Wakolbinger, 1986; Millet et al., 1989; Da Prato, 2006; Da Prato and Zabczyk, 2014), SGMs have been extended to operate directly in Hilbert function spaces (Kerrigan et al., 2022; Lim et al., 2023; Franzese et al., 2024; Pidstrigach et al., 2023; Hagemann et al., 2023; Bond-Taylor and Willcocks, 2023; Lim et al., 2024). Some works have started employing infinite-dimensional SGMs to solve inverse problems, providing a discretization-invariant numerical platform for exploring the posterior (Pidstrigach et al., 2023; Baldassari et al., 2024; Hosseini et al., 2023). However, the theoretical guarantees provided by these works require the inverse problem to be linear, whereas many interesting inverse problems are *nonlinear*, like those arising in electrical impedance tomography (Calderón, 2006; Borcea, 2002; Uhlmann, 2009), data assimilation (Law et al., 2015), photo-acoustic tomography (Bal and Ren, 2011; Bal and Uhlmann, 2010), boundary rigidity (Kachalov et al., 2001), and groundwater flow (Dashti and Stuart, 2011).

In this work, we take a first step towards bridging this gap and utilize an *infinite-dimensional unconditional score model* as a prior in a Langevin-type MCMC algorithm, providing theoretical guarantees for its convergence to the true posterior of function-space nonlinear inverse problems. In doing so, we extend the theoretical setup of Sun et al. (2024) to Hilbert function spaces, presenting a convergence analysis with error bounds that are *dimension-free*. The main feature of our analysis, similar to Sun et al. (2024), is that it is fully compatible with the joint presence of potentially non-log-concave likelihoods (making it suitable for nonlinear inverse problems), imperfect score networks, and weighted annealing. Most importantly, we prove that controlling the approximation error of the score is essential for ensuring convergence and that *modifying the standard Langevin-type MCMC algorithm through the selection of an appropriate preconditioner is necessary*. More precisely, our analysis shows how the control over the score approximation error dictates the design of the preconditioner—an aspect unique to the infinite-dimensional setting.

## 1.1 RELATED WORK

Since we aim to extend the theoretical setup of Sun et al. (2024) into infinite dimensions, while addressing the theoretical questions posed by Stuart (2010) on the challenges of functions-space

Bayesian inference, our work combines elements from three contemporary research areas: MCMC methods for functions, Bayesian nonlinear inverse problems, and SGMs.

There exists a large body of literature on infinite-dimensional score-based MCMC algorithms (Beskos et al., 2017; Wallin and Vadlamani, 2018; Durmus and Moulines, 2019; 2017; Dalalyan, 2017; Hairer et al., 2014; Cotter et al., 2013; Cui et al., 2016; 2024; Beskos et al., 2018; Morzfeld et al., 2019; Muzellec et al., 2022; Beskos et al., 2008). However, most of these works precede the recent wave of papers on SGMs. The main inspiration behind our theory is the non-asymptotic stationary convergence analysis recently developed in the finite-dimensional setting by Sun et al. (2024) for a method known by various names, such as the plug-and-play unadjusted Langevin algorithm or *plug-and-play Monte Carlo* (PMC-RED), with the latter making the connection to regularization-by-denoising algorithms (Reehorst and Schniter, 2018; Romano et al., 2017) explicit. This method, closely related to stochastic gradient Langevin dynamics (Welling and Teh, 2011), employs a plug-and-play approach within an MCMC scheme. Specifically, it aims to learn an approximation of the prior density through a denoising algorithm while keeping an explicit likelihood, in the same spirit as fixed-point algorithms (Buzzard et al., 2018). While similar methods have been used frequently in the past (Venkatakrishnan et al., 2013; Alain and Bengio, 2014; Guo et al., 2019; Kadkhodaie and Simoncelli, 2021), a general proof of convergence in the context of stochastic Bayesian algorithms was only recently proposed by Laumont et al. (2022). Sun et al. (2024) rely on weaker conditions, and their analysis is compatible with the joint presence of potentially non-log-concave likelihoods, imperfect score networks, and weighted annealing. Unfortunately, their convergence bound is not dimension-free and becomes uninformative in the limit of infinite dimensions. In our work, we fill this gap by *carrying out the convergence analysis in function spaces*.

Using MCMC methods that provably sample from a non-log-concave posterior distribution, especially in function spaces, is notoriously challenging since it results in a high-dimensional, non-convex optimization problem. Recently, a series of rigorous mathematical papers, mostly by Richard Nickl and his collaborators, have approached nonlinear inverse problems within a probabilistic framework (Nickl and Wang, 2022; Nickl and Söhl, 2019; Nickl, 2020; Abraham, 2019; Furuya et al., 2024; Giordano and Nickl, 2020; Bohr and Nickl, 2021; Paternain et al., 2012; Monard et al., 2021a; Bonito et al., 2017; Nickl and Paternain, 2022; Vershynin, 2018; Nickl et al., 2020; Nickl and Söhl, 2017; Monard et al., 2021b; Spokoiny, 2019); see Nickl (2023a) for an overview. The general idea is to provide a set of assumptions for the forward model to mitigate the non-log-concavity of the posterior. The main concerns of these works are ensuring *statistical consistency*, i.e., that the posterior concentrates most of its mass around the actual parameter that generated the data, and *computability*. For the former, the global stability of the inverse problem appears to be a sufficient condition. While we have not addressed this in our work, it can be imposed by restricting the family of nonlinear inverse problems under consideration, thus without changing the essence of our convergence analysis. A stronger assumption—local gradient stability of the forward map—is crucial for computability, as it ensures local log-concavity of the posterior. This implies that if a Markov chain is initialized in such a local region, proving convergence and fast mixing time of the sampling procedure becomes easier. We discuss the challenges related to the computational complexity of Langevin-type MCMC algorithms in the Discussion and Conclusion section; it's worth mentioning, however, that in our work we focus only on theoretical convergence, even though our setup, being compatible with a weighted annealing schedule, provides a heuristic to speed up the mixing of the Markov chain, similar to Song and Ermon (2019) and Sun et al. (2024).

In our convergence analysis, the learned score plays a key role. Among the theoretical frameworks defining SGMs in infinite dimensions, we consider the one by Pidstrigach et al. (2023) and Baldassari et al. (2024) for continuous-time diffusion models. An important contribution of our work is that we show not only that the obtained convergence bound explicitly depends on the $H$-accuracy of the approximated score—where $H$ is the infinite-dimensional separable Hilbert space in which the inverse problem is defined—but also that the control we have over the score approximation error plays a key role in designing the MCMC sampler, particularly in *introducing the preconditioning operator that ensures convergence in function spaces*. The idea of modifying an MCMC sampler with a preconditioner in the infinite-dimensional setting is not new (Hairer et al., 2007); however, it is novel in the context of SGMs. In fact, we not only prove convergence with *imperfect scores*, similar to Sun et al. (2024), but we also characterize the parameters of the preconditioner with respect to the strength of the control over the score approximation error.

## 1.2 OUR CONTRIBUTION

In this work, we provide *theoretical guarantees for the convergence of a Hilbert space Langevin-type MCMC algorithm that incorporates infinite-dimensional unconditional score models and samples from the posterior of nonlinear inverse problems*. The main contributions are as follows:

- We study the extension to infinite dimensions of the posterior sampler defined by Sun et al. (2024) by utilizing infinite-dimensional SGMs as expressive learning-based priors within a Hilbert space Langevin-type MCMC scheme (Section 4).

- In doing so, we build upon the non-asymptotic stationary convergence analysis of Sun et al. (2024). We prove that the infinite-dimensional algorithm converges to the posterior under possibly non-log-concave likelihoods and imperfect scores. The obtained convergence bound is dimension-free and depends on the score approximation error (Theorem 1).

- The role of the score approximation error is explored in detail, as ensuring the convergence of the algorithm in Sun et al. (2024) in infinite dimensions requires the use of an appropriate preconditioner whose parameters depend on the strength of the control over the error. This aspect is quantified (Remark 7).

Section 5 concludes with a discussion on the challenges related to learning the score and computational complexity of Langevin-type MCMC methods, drawing connections to Song and Ermon (2019) and Nickl (2023b).

## 2 BACKGROUND

### 2.1 THE BAYESIAN APPROACH TO INVERSE PROBLEMS

We consider the possibly nonlinear inverse problem

$$\mathbf{y} = \mathcal{A}(X_0) + \mathbf{b}, \tag{1}$$

where the unknown parameter $X_0$ is modeled as an $H$-valued random variable and $H$ is an infinite-dimensional separable Hilbert space, $\mathcal{A} : H \to \mathbb{R}^N$ is the measurement operator, and $\mathbf{b}$ is the noise term with a given density $\rho$ with respect to the Lebesgue measure over $\mathbb{R}^N$. We assume to have some prior knowledge about the distribution of $X_0$ before any measurements are taken. This knowledge is encoded in a given prior measure $\mu_0$. The solution to (1) is then represented by the conditional probability measure of $X_0|\mathbf{y} \sim \mu^{\mathbf{y}}$, which is typically referred to as the posterior (Stuart, 2010). If $\mathbb{E}_{\mu_0}[\rho(\mathbf{y} - \mathcal{A}(X_0))] < +\infty$, which is the case for instance when the density $\rho$ is bounded (such as a multivariate Gaussian $\mathcal{N}(0, \mathbf{\Gamma})$), then $\mu^{\mathbf{y}}$ is absolutely continuous with respect to $\mu_0$ (we write $\mu^{\mathbf{y}} \ll \mu_0$) and its Radon-Nikodym derivative is given by

$$\frac{d\mu^{\mathbf{y}}}{d\mu_0}(X) = \frac{1}{Z(\mathbf{y})}\exp(-\Phi_0(X; \mathbf{y})), \tag{2}$$

where $\Phi_0(X; \mathbf{y}) := -\log\big(\rho(\mathbf{y} - \mathcal{A}(X))\big)$ is the negative log-likelihood. Explicitly characterizing $\mu^{\mathbf{y}}$ in (2) is challenging, particularly in high dimensions, due to the intractable normalizing constant $Z(\mathbf{y}) := \int_H \exp(-\Phi_0(X; \mathbf{y}))d\mu_0(X)$. Popular methods for exploring the posterior, such as Langevin-type MCMC algorithms, aim to generate samples distributed approximately according to $\mu^{\mathbf{y}}$. As anticipated in the Introduction section, we propose to extend one such algorithms to infinite dimensions: the *plug-and-play Monte Carlo method* (PMC-RED) proposed by Sun et al. (2024).

### 2.2 FORMULATION OF PMC-RED

For the reader's convenience, we will now review the formulation of PMC-RED proposed by Sun et al. (2024) for sampling the posterior of a possibly nonlinear imaging inverse problem in finite dimensions, $\mathbf{y} = \mathbf{A}(\mathbf{x}) + \mathbf{e}$, with $\mathbf{A} : \mathbb{R}^n \to \mathbb{R}^m$ and $\mathbf{e} \sim \mathcal{N}(0, \sigma^2\mathbf{I})$. As the reader will notice, this method resembles the well-known stochastic gradient Langevin dynamics (Welling and Teh, 2011), with the main difference being that the prior is replaced by the score network of a *smoothed prior*, which provides additional regularity in computing the gradient.

PMC-RED is built on the fusion of traditional regularization-by-denoising (RED) algorithms (Reehorst and Schniter, 2018; Romano et al., 2017) and score-based generative modelling (Song and

Ermon, 2019; Song et al., 2020; Ho et al., 2020). It incorporates expressive score-based generative priors in a plug-and-play fashion (Venkatakrishnan et al., 2013; Alain and Bengio, 2014; Guo et al., 2019; Kadkhodaie and Simoncelli, 2021) for conducting provable posterior sampling. Given an initial state $\mathbf{x}_0 \in \mathbb{R}^n$, PCM-RED is defined as the following recursion

$$\mathbf{x}_{k+1} = \mathbf{x}_k - \gamma\big(\nabla g(\mathbf{x}_k) - \mathbf{S}_\theta(\mathbf{x}_k, \sigma)\big) + \sqrt{2\gamma}\mathbf{Z}_k, \tag{3}$$

where $\mathbf{Z}_k = \int_k^{k+1} d\mathbf{W}_t$ follows the $m$-dimensional i.i.d normal distribution, $\{\mathbf{W}_t\}_{t\geq 0}$ represents the $m$-dimensional Brownian motion, $\gamma > 0$ denotes the step-size, $g$ is the negative log-likelihood, and $\mathbf{S}_\theta(\mathbf{x}_k, \sigma) \approx \nabla \log p_\sigma(\mathbf{x}_k)$ is the score network for $p_\sigma$, a smoothed prior with $\nabla \log p_\sigma \to \nabla \log p$ as $\sigma \to 0$. As we mentioned, a motivation for using $p_\sigma$ is that $p$ may be non-differentiable, precluding the use of algorithms such as gradient descent for maximum a posteriori (MAP) estimation. This motivates the application of proximal methods (Beck and Teboulle, 2009a; Boyd et al., 2011) like RED (Beck and Teboulle, 2009b). Interestingly, Sun et al. (2024) notice that since the gradient-flow ODE links RED to the Langevin diffusion described by the SDE

$$d\mathbf{x}_t = \big(\nabla \log p(\mathbf{x}_t) - \nabla g(\mathbf{x}_t)\big)dt + \sqrt{2}d\mathbf{W}_t,$$

one can interpret (3) as a parallel MCMC algorithm of RED for posterior sampling—in this sense, PMC-RED is conceptually equivalent to the plug-and-play unadjusted Langevin algorithm studied by Laumont et al. (2022). The reason we focus on PMC-RED instead of other similar methods is that Sun et al. (2024) provide a convergence analysis that is compatible with the joint presence of potentially non-log-concave likelihoods, imperfect scores, and weighted annealing. Unfortunately, the obtained convergence bound depends on the dimension of the problem, and thus becomes uninformative in infinite dimensions. To address this issue, in Section 4 we carry out the convergence analysis directly in Hilbert (function) spaces.

## 3 SCORE-BASED GENERATIVE PRIORS IN HILBERT SPACE

In (3), the score network $\mathbf{S}_\theta(\mathbf{x}, \sigma)$ approximates $\nabla \log p_\sigma(\mathbf{x})$. As mentioned above, $p_\sigma$ refers to the smoothed prior, here being a distribution with respect to the Lebesgue measure. In infinite dimensions, however, there is no natural analogue of the Lebesgue measure; $p_\sigma$ is no longer well defined (Da Prato, 2006). To extend PMC-RED to infinite dimensions we then need to define the infinite-dimensional score function that will replace $\nabla \log p_\sigma(\mathbf{x})$. Subsequently, we will show in Section 4 that this allows us to approximately sample from $\mu^\mathbf{y}$ in the infinite-dimensional setting.

Let $C_{\mu_0} : H \to H$ be a trace class, positive-definite, symmetric covariance operator. Here and throughout the paper, we assume that the prior $\mu_0$ that we want to learn from data to perform Bayesian inference in (1) is the Gaussian measure $\mu_0 = \mathcal{N}(0, C_{\mu_0})$, though our analysis can be easily generalized to other classes of priors, e.g. priors given as a density with respect to a Gaussian. To define the score function in infinite dimensions, we follow the continuous-time approach outlined in Pidstrigach et al. (2023) and Baldassari et al. (2024). Let $C : H \to H$ be a trace class, positive-definite, symmetric covariance operator. Let $\{W_t\}_{t\geq 0}$ be a Wiener process on $H$. Denote by $X_\tau$ the diffusion at time $\tau$ of a prior sample $X_0 \sim \mu_0$:

$$X_\tau := e^{-\tau/2}X_0 + \int_0^\tau e^{-(\tau-s)/2}\sqrt{C}dW_s.$$

$X_\tau$ evolves towards the Gaussian measure $\mathcal{N}(0, C)$ as $\tau \to \infty$ according to the SDE

$$dX_\tau = -\frac{1}{2}X_\tau d\tau + \sqrt{C}dW_\tau, \qquad X_0 \sim \mu_0. \tag{4}$$

The score function in infinite dimensions is defined as follows:

**Definition 1.** *The score function enabling the time-reversal of* (4) *is defined for $x \in H$ as*

$$S(\tau, x; \mu_0) := -(1 - e^{-\tau})^{-1}(x - e^{-\tau/2}\mathbb{E}[X_0|X_\tau = x]). \tag{5}$$

**Remark 1.** *The neural network $S_\theta(\tau, x; \mu_0)$ that approximates the true score minimizes the denoising score matching loss in infinite dimensions:*

$$\mathbb{E}_{x_0 \sim \mathcal{L}(X_0), x_\tau \sim \mathcal{L}(X_\tau|X_0=x_0)}[\|S_\theta(\tau, x_\tau; \mu_0) - (1 - e^{-\tau})^{-1}(x_\tau - e^{-\tau/2}x_0)\|_H^2],$$

*where $\mathcal{L}(X_0)$ and $\mathcal{L}(X_\tau|X_0 = x_0)$ denote the law of $X_0$ and $X_\tau|X_0 = x_0$, respectively.*

**Remark 2.** $S_\theta(\tau, x; \mu_0)$ *for some $\tau > 0$ is what will replace the approximation of $\nabla \log p_\sigma$ in the finite-dimensional case* (3).

We will now make an assumption, often employed in infinite dimensions on $C$ and $C_{\mu_0}$.

**Assumption 1.** *We assume that $C_{\mu_0}, C : H \to H$ have the same basis of eigenvectors $(e_j)$ and that $C_{\mu_0} e_j = \mu_{0j} e_j$, $Ce_j = \lambda_j e_j$ for every $j$, with $\lambda_j / \mu_{0j} < +\infty$.*

**Remark 3.** *If $C = C_{\mu_0}$ or if $C$ is very close to $C_{\mu_0}$, in the sense that $C^{-1/2} C_{\mu_0} C^{-1/2} - I$ is Hilbert-Schmidt (Da Prato and Zabczyk, 2014), then $\sup \lambda_j / \mu_{0j} < +\infty$.*

The proposition below has been proved by Baldassari et al. (2024). We will reproduce the proof for the reader's convenience in Appendix A.

**Proposition 1.** *Let Assumption 1 hold. Then $S(\tau, x; \mu_0) = -\sum_j \frac{e^\tau p_0^{(j)}}{1+(e^\tau-1)p_0^{(j)}} x^{(j)} e_j = -CC_\tau^{-1} x$, where $x^{(j)} := \langle x, e_j \rangle$, $p_0^{(j)} := \frac{\lambda_j}{\mu_{0j}}$, and $C_\tau := e^{-\tau} C_{\mu_0} + (1 - e^{-\tau}) C$.*

Throughout the paper, we assume the situation described in Remark 3. In particular, the condition for the spectral norm $\|CC_{\mu_0}^{-1}\| = \sup_j \lambda_j / \mu_{0j} < +\infty$ is needed to ensure the following result, which will be useful in the proof of Theorem 1

**Corollary 1.** *By Proposition 1, $CC_{\mu_0}^{-1} x + S(\tau, x; \mu_0) = (e^\tau - 1) \sum_j^\infty \frac{p_0^{(j)}-1}{1+(e^\tau-1)p_0^{(j)}} p_0^{(j)} x^{(j)} e_j$. Moreover, the following inequality for the so-called* score mismatch error *holds:*

$$\|CC_{\mu_0}^{-1} x + S(\tau, x; \mu_0)\|_H^2 \le (e^\tau - 1)^2 (\|CC_{\mu_0}^{-1}\| + 1)^2 \|CC_{\mu_0}^{-1}\|^2 \|x\|_H^2.$$

**Remark 4.** *Similar results to Corollary 1 can be derived when $\mu_0$ is given as a density with respect to a Gaussian, $d\mu_0(x) = \exp(\Psi(x)) d\mathcal{N}(0, C_{\mu_0})$; see Theorem 3 of Pidstrigach et al. (2023).*

## 4 HILBERT SPACE MCMC WITH SCORE-BASED PRIORS

We now utilize the score network $S_\theta(\tau, x; \mu_0)$ introduced in Remark 1 as a learning-based prior and modify the Langevin dynamics of PMC-RED so that it can operate directly in function spaces. The infinite-dimensional version of (3) is then defined as follows:

$$X_{k+1} = X_k - \gamma \big( -C^{\alpha-1} S_\theta(\tau, X_k; \mu_0) - C^\alpha \nabla_{X_k} \log(\rho(\mathbf{y} - \mathcal{A}(X_k))) \big) + (2\gamma)^{\frac{1}{2}} Z_k, \quad (6)$$

where $\gamma > 0$ denotes the step-size, $\alpha > 0$ is a constant that will be chosen later, and $Z_k = \int_k^{k+1} C^{\frac{\alpha}{2}} dW_t$ denotes the i.i.d Gaussian variables with mean zero and covariance operator $C^\alpha$. Our main theoretical result, as summarized later in Theorem 1, presents a convergence analysis of (6), demonstrating that when $\tau$ is sufficiently small and $S_\theta$ provides a good approximation of the true score of the prior, it generates samples distributed approximately according to the true posterior. Finally, as the reader will have immediately noticed, (6) differs from PMC-RED due to the presence of a preconditioner $C^\alpha$. The role of $C$ and $\alpha$ will be explored thoroughly in this section, where we provide a detailed convergence analysis of the sampler defined in (6).

### 4.1 MEASURE-THEORETIC DEFINITIONS OF THE KL DIVERGENCE AND THE FISHER INFORMATION

Before introducing our convergence theorem, we need to introduce analogues of the metrics appearing in Theorem 1 of Sun et al. (2024)—namely, the Kullback-Leibler (KL) divergence and the relative Fisher information (FI)—that are compatible with the infinite-dimensional setting of our paper. Since, as we mentioned, there is no natural analogue of the Lebesgue measure in infinite-dimensional spaces, we will adopt a measure-theoretic definition of the KL divergence (Ambrosio et al., 2005):

$$\text{KL}(\nu || \mu) := \int_H \log \frac{d\nu}{d\mu}(X) d\nu(X)$$

if $\nu \ll \mu$, where $d\nu/d\mu$ refers to the Radon-Nikodym derivative; this quantity is set to infinity if $\nu$ is not absolutely continuous with respect to $\mu$. In our convergence theorem, we will employ

$$\int_H \left\| C^{\frac{\alpha}{2}} \nabla_X \log \frac{d\nu}{d\mu}(X) \right\|_H^2 d\nu(X) \quad (7)$$

as a criterion to assess similarity between measures. As for the measure-theoretic KL divergence, we set (7) to infinity if $\nu$ is not absolutely continuous with respect to $\mu$. It is straightforward to see that, if (7) is zero, then $\nu$ and $\mu$ are equal $\nu$-almost surely.

## 4.2 THEORETICAL CONVERGENCE ANALYSIS

We aim to study the convergence of

$$\int_H \left\| C^{\frac{\alpha}{2}} \nabla_X \log \frac{d\nu_t}{d\mu^{\mathbf{y}}}(X) \right\|_H^2 d\nu_t(X), \tag{8}$$

where $\{\nu_t\}_{t\geq 0}$ represents a continuous interpolation of the probability measures generated by (6)

$$X_t = X_{k\gamma} + (t - k\gamma)\big(C^{\alpha-1}S_\theta(\tau, X_{k\gamma}; \mu_0) + C^\alpha \nabla_{X_{k\gamma}} \log(\rho(\mathbf{y} - \mathcal{A}(X_{k\gamma})))\big) + 2^{\frac{1}{2}}C^{\frac{\alpha}{2}}(W_t - W_{k\gamma}) \tag{9}$$

for $t \in [k\gamma, (k+1)\gamma]$, with initial state $X_0 \sim \nu_0$, where $\gamma > 0$ is the step-size and $S_\theta$ is a neural network approximating the score defined in (5). To prove the convergence of (8), we will use the following assumptions.

**Assumption 2.** $\nabla_X \Phi_0$ *is continuously differentiable and globally Lipschitz; for any* $X_1, X_2 \in H$:

$$\|\nabla \Phi_0(X_1) - \nabla \Phi_0(X_2)\|_H \leq L_{\Phi_0}\|X_1 - X_2\|_H.$$

**Remark 5.** *Note that Assumption 2 does not assume the log-concavity of the likelihood, meaning that our analysis is compatible with nonlinear inverse problems.*

**Assumption 3.** *For any* $\tau > 0$*, the score network* $S_\theta(\tau, X; \mu_0)$ *approximating* (5) *is Lipschitz continuous with* $L_\tau > 0$ *for any* $X_1, X_2 \in H$:

$$\|S_\theta(\tau, X_1; \mu_0) - S_\theta(\tau, X_2; \mu_0)\|_H \leq L_\tau \|X_1 - X_2\|_H. \tag{10}$$

*Moreover,* $S_\theta(\tau, X; \mu_0)$ *has a bounded error* $\epsilon_\tau < \infty$ *for every* $X \in H$:

$$\|S_\theta(\tau, X; \mu_0) - S(\tau, X; \mu_0)\|_H \leq \epsilon_\tau. \tag{11}$$

**Assumption 4.** *The forward operator* $\mathcal{A}$ *depends only on* $P^{D_0}(X)$ *for some* $D_0 > 0$*. Moreover, we assume that the* $\nu_0$ *introduced in* (9) *can be factorised as follows*

$$\nu_0(X) = \nu_0^{D_0}(X^{D_0}) \prod_{j=D_0+1}^{\infty} \nu_0^{(j)}(X^{(j)}),$$

*where the superscript* $D_0$ *in* $X^{D_0}$ *refers to the orthogonal projection* $P^{D_0}$ *of the* $H$*-valued random variable* $X$ *onto the linear span of the first* $D_0$ *eigenvectors* $(e_j)$ *of* $C$*,* $\nu_0^{D_0} := P^{D_0}_\# \nu_0$*, and* $\nu_0^{(j)}$ *is the density of* $X^{(j)} := \langle X, e_j \rangle$*; see Appendix B.1 for details on the notation. We also assume that*

$$\mu^{\mathbf{y}}(X) = (\mu^{\mathbf{y}})^{D_0}(X^{D_0}) \prod_{j=D_0+1}^{\infty} (\mu^{\mathbf{y}})^{(j)}(X^{(j)}).$$

**Remark 6.** *Assumption 4 implies that the algorithm does not explicitly depend on the articulation of the subspace associated with the first* $D_0$ *modes. Thus, the essential aspect of the assumption is that only a finite number of modes contributes to the observations, which is quite realistic from an applications point of view. Moreover, the error bound in Theorem 1 will not depend on* $D_0$*, ensuring the robustness of the convergence analysis with respect to increasing* $D_0$*, which is crucial in an infinite-dimensional setting.*

Now that we have listed the main assumptions for our convergence analysis, we are ready to state our main result, Theorem 1, where we establish an explicit bound for (8), which resembles that for PMC-RED in Sun et al. (2024), with the main difference being that ours does not diverge as the dimension of the problem goes to infinity. Additionally, our proof rigorously quantifies the relationship between $C$, $\alpha$, and the score approximation error—an aspect unique to the infinite-dimensional setting.

**Theorem 1** (Convergence bound on Hilbert spaces). *Let Assumptions 1–4 hold. Denote by $\{\nu_t\}_{t \geq 0}$ a continuous interpolation of the probability measures generated by (6):*

$$X_t = X_{k\gamma} + (t - k\gamma)\left(C^{\alpha-1}S_\theta(\tau, X_{k\gamma}; \mu_0) + C^\alpha \nabla_{X_{k\gamma}}\log(\rho(\mathbf{y} - \mathcal{A}(X_{k\gamma})))\right) + 2^{\frac{1}{2}}C^{\frac{\alpha}{2}}(W_t - W_{k\gamma})$$

*for $t \in [k\gamma, (k+1)\gamma]$, with initial state $X_0 \sim \nu_0$, where $\gamma > 0$ is the step-size, $S_\theta$ is a neural network approximating the score defined in (5), and $\{W_t\}_{t \geq 0}$ is a Wiener process on $H$ independent of $X_t$. For $\alpha \geq 2$ and $\gamma \in \left(0, \frac{1}{\sqrt{128}\mathrm{Tr}(C^\alpha)L_\mathcal{G}}\right]$, we have*

$$\frac{1}{N\gamma}\int_0^{N\gamma}\left(\int\left\|C^{\frac{\alpha}{2}}\nabla_X\log\frac{d\nu_t}{d\mu^{\mathbf{y}}}(X)\right\|_H^2 d\nu_t(X)\right)dt$$

$$\leq \frac{4\mathrm{KL}(\nu_0\|\mu^{\mathbf{y}})}{N\gamma} + \left(\frac{32\sqrt{2}}{3} + 64\right)\mathrm{Tr}(C^\alpha)L_\mathcal{G}^2\gamma + \underbrace{\frac{52}{3}K^2}_{\substack{\text{Score Mismatch}\\\text{Error}}}\tau^2 + \underbrace{\frac{52}{3}\|C\|^{\alpha-2}}_{\substack{\text{Score Approximation}\\\text{Error}}}\epsilon_\tau^2,$$

*where $L_\mathcal{G}^2 := \|C\|^{\alpha-2}L_\tau^2 + \|C\|^\alpha L_{\Phi_0}^2$, $K^2 := \|C\|^{\alpha-2}(\|CC_{\mu_0}^{-1}\| + 1)^2\|CC_{\mu_0}^{-1}\|^2\sup_{t\in[0,N\gamma]}\mathbb{E}[\|X_t\|_H^2]$, $\|\cdot\|$ denotes the spectral norm, and $N > 0$ is the total number of iterations.*

*Proof.* (Sketch, the full proof can be found in Appendix B) We define the stochastic process

$$X_t := X_0 + t\left(C^{\alpha-1}S(\tau, X_0; \mu_0) + C^\alpha\nabla_{X_0}\log(\rho(\mathbf{y} - \mathcal{A}(X_0)))\right) + 2^{\frac{1}{2}}C^{\frac{\alpha}{2}}W_t, \quad X_0 \sim \nu_0.$$

We derive the evolution equation for $\nu_t$ (the probability measure of $X_t$) and plug it into the time derivative formula for $\mathrm{KL}(\nu_t\|\mu^{\mathbf{y}})$.

We derive a bound relating $\int_H\left\|C^{\frac{\alpha}{2}}\nabla_X\log\frac{d\nu_t}{d\mu^{\mathbf{y}}}(X)\right\|_H^2 d\nu_t(X)$ and the expected square $H$-norm $\mathbb{E}\left[\|C^{\frac{\alpha}{2}-1}S(\tau, X; \mu_0) + C^{\frac{\alpha}{2}}\nabla_X\log(\rho(\mathbf{y} - \mathcal{A}(X)))\|_H^2\right]$.

We construct a linear interpolation of (6), make use of the aforementioned bounds and Assumptions 1–4, and integrate the time derivative of the KL divergence over the interval $[k\gamma, (k+1)\gamma]$ to obtain a convergence bound that is dimension-free and depends explicitly on the score mismatch and the network approximation errors. $\square$

**Corollary 2** (Stationarity). *Let $\alpha \geq 2$. If $\gamma$, $\tau$, and $\epsilon_\tau$ are sufficiently small, then $\nu_t$ converges to $\mu^{\mathbf{y}}$ in terms of the averaged measure-theoretic FI (7) at the rate of $\mathcal{O}\left(\frac{1}{N}\right)$.*

**Remark 7.** *The interplay between $C$, $\alpha$, and the imperfect score networks is a novel aspect of our analysis and can be rigorously quantified. Indeed, if we have a better control on the score approximation error, such as $\|C^{-\beta}(S_\theta - S)\|_H \leq \epsilon_{\beta,\tau}$ for some $\beta \geq 0$, then our proof of Theorem 1 shows that the score approximation error term $\|C\|^{\alpha-2}\epsilon_\tau^2$ can be replaced by $\|C\|^{\alpha-2+2\beta}\epsilon_{\beta,\tau}^2$.*

We conclude this section by briefly discussing *weighted annealing*, a well-known heuristic to mitigate mode collapse and accelerate the sampling speed of Langevin MCMC algorithms (Song and Ermon, 2019; Kirkpatrick et al., 1983; Neal, 2001). It consists of replacing $S_\theta(\tau, X_k; \mu_0)$ in (6) by $\eta_k S_\theta(\tau_k, X_k; \mu_0)$, where $(\eta_k)$ and $(\tau_k)$ decay from large initial values to 1 and almost 0, respectively. Following our proof of Theorem 1, one can show that weighted annealing will not introduce extra error influencing the convergence accuracy in the infinite-dimensional case. Additional assumptions, however, will be needed, such as that the output of the score network $S_\theta(\tau, X; \mu_0)$ is bounded in $H$-norm and the Lipschitz constant of the true score is not exploding, as this is necessary for the existence of $\sup\{L_{\tau_k}\}_{k=0}^{N-1}$ as $N \to \infty$, where $L_{\tau_k}$ denotes the Lipschitz constant of $S_\theta(\tau_k, X_k; \mu_0)$.

## 5    DISCUSSION AND CONCLUSION

In this work, we address the concerns raised by Stuart (2010), who emphasized the importance of using algorithms specifically designed for the infinite-dimensional setting when dealing with intrinsically infinite-dimensional objects in the context of inverse problems. We extend one of the methods commonly referred to as score-based generative models (SGMs) to Hilbert spaces, where the learned score is used as a prior in a Langevin-type MCMC algorithm for posterior sampling (Sun et al.,

2024). By leveraging the recently developed infinite-dimensional framework for SGMs, we provide theoretical guarantees for the convergence of the MCMC sampler that uses the infinite-dimensional unconditional score as a prior. Our analysis, conducted in the challenging context of nonlinear Bayesian inverse problems, shows that controlling the approximation error of the score is not only essential for ensuring convergence but also that modifying the Langevin MCMC algorithm through the selection of an appropriate preconditioner is necessary. Our analysis shows how the control over the score approximation error influences the design of the preconditioner—an aspect unique to the infinite-dimensional setting.

Despite the rigor of our convergence analysis, we anticipate that practical challenges common to most Langevin-type MCMC algorithms, as described in Section 3 of Song and Ermon (2019), particularly in learning the score in low-density regions and the mixing times of the Langevin MCMC algorithm, will carry over to the infinite-dimensional setting. For the former, it is known that to accurately sample from the posterior distribution, the SGM must precisely estimate the scores for both the initial point in the MCMC chain and all points during the burn-in phase. However, when $\tau$ is small, since there is no guarantee that the MCMC chain explores the high-probability regions of the prior during burn-in, the estimated scores might be inaccurate, possibly preventing the chain from converging to the true posterior. One possible heuristic for addressing this issue involves adopting a weighted annealed schedule, as suggested by Song and Ermon (2019) and Sun et al. (2024), among others. Another challenge comes from the non-convexity originating from the nonlinearity of the inverse problem. If not handled properly, Langevin-type MCMC algorithms are known to converge slowly or, worse, get stuck in local minima. Nickl (2023a) provides algorithmic guarantees, but they rely on strong assumptions about the forward model. Our work addresses the theoretical convergence under weaker assumptions. Moreover, weighted annealing has shown promising results in addressing issues related to mixing time and local minima. In offering a theoretical foundation to show that Hilbert space Langevin MCMC samplers with score-based priors are provably convergent, we leave to future work the derivation of algorithmic strategies to overcome the challenges outlined above.

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

## A   PROOF OF PROPOSITION 1

We define $X_\tau^{(j)} = \langle X_\tau, e_j \rangle$ and $S^{(j)}(\tau, x; \mu_0) = \langle S(\tau, x; \mu_0), e_j \rangle$. We then have

$$dX_\tau^{(j)} = -\frac{1}{2} X_\tau^{(j)} d\tau + \sqrt{\lambda_j} W_\tau^{(j)}.$$

Since $C_{\mu_0}$ and $C$ have the same basis of eigenfunctions, the system of modes diagonalizes so that the $X_\tau^{(j)}$ processes are independent for different $j$ modes. Thus we have

$$X_0^{(j)} = \sqrt{\mu_{0j}}\eta_0^{(j)}, \quad X_\tau^{(j)} = X_0^{(j)} e^{-\tau/2} + \sqrt{\lambda_j(1 - e^{-\tau})}\eta_1^{(j)}$$

for $\eta_i^{(j)}$ independent standard Gaussian random variables. We seek

$$x_0^{(j)} = \mathbb{E}[X_0^{(j)} | X_\tau^{(j)} = x^{(j)}],$$

where $x_0^{(j)} = ax^{(j)}$ with $a$ solving

$$\mathbb{E}[(aX_\tau^{(j)} - X_0^{(j)})X_\tau^{(j)}] = 0,$$

which gives

$$a = \frac{e^{\tau/2}}{1 + (e^\tau - 1)p_0^{(j)}}$$

for

$$p_0^{(j)} = \frac{\lambda_j}{\mu_{0j}}.$$

Since also the time-reversed system diagonalizes, we have

$$S^{(j)}(\tau, x; \mu_0) = S^{(j)}(\tau, x^{(j)}; \mu_0) = -\left(\frac{e^\tau p_0^{(j)}}{1 + (e^\tau - 1)p_0^{(j)}}\right) x^{(j)}.$$

## B   PROOF OF THE CONVERGENCE THEOREM

### B.1   FINITE-DIMENSIONAL PROJECTION

Denote by $(e_j)$ the orthonormal basis of eigenvectors of a trace class, positive-definite, symmetric covariance operator $C$.

**Definition 2.** *Define the linear span of the first $D$ eigenvectors as*

$$H^D := \left\{ \sum_{j=1}^D f_j e_j | f_1, \ldots, f_D \in \mathbb{R} \right\} \subset H.$$

*Define $H^{D+1:\infty}$ such that $H = H^D \otimes H^{D+1:\infty}$.*

**Definition 3.** *Let $P^D : H \to H^D$ be the orthogonal projection onto $H^D$. If we write an element $f$ of $H$ as*

$$f = \sum_{j=1}^{\infty} \langle f, e_j \rangle e_j,$$

*$P^D$ is equivalent to restricting $f$ to its first $D$ coefficients:*

$$P^D f = \sum_{j=1}^{D} \langle f, e_j \rangle e_j.$$

**Definition 4.** *The push-forward $P^D_{\#} \mu$ of $\mu$ under $P^D$ is denoted by*

$$\mu^D := P^D_{\#} \mu, \qquad where \quad P^D_{\#} \mu(A) = \mu((P^D)^{-1}(A)).$$

## B.2 Useful Results on Measure Theory

Here we review some basic measure-theoretic tools needed in the proof of Theorem 1.

**Theorem 2** (Disintegration (Ambrosio et al., 2005)). *Let $\mathcal{P}(H)$ be the family of all Borel probability measures on $H$. Let $H,Y$ be Radon separable metric spaces, $\mu \in \mathcal{P}(H)$, and $\pi : H \to Y$ a Borel map. Then there exists a $\pi_{\#}\mu$-a.e. uniquely determined Borel family of probability measures $\{\mu_y\}_{y \in Y} \subset \mathcal{P}(H)$ such that*

$$\mu_y(H \setminus \pi^{-1}(y)) = 0 \qquad for \ \pi_{\#}\mu\text{-}a.e. \ y \in Y$$

*and*

$$\int_H f(x)d\mu(x) = \int_Y \left( \int_{\pi^{-1}(y)} f(x)d\mu_y(x) \right) d\pi_{\#}\mu(y)$$

*for every Borel map $f : H \to [0, \infty]$. In particular, when $H = H^D \times H^{D+1:\infty}$, $Y = H^D$, and $\pi = P^D$ (hence $\pi_{\#}\mu = \mu^D$), we can identify $\pi^{-1}(x^D)$ with $H^{D+1:\infty}$ and find a Borel family of probability measures $\{\mu_{x^D}\}_{x^D \in H^D}$ such that*

$$\mu_{x^D}(H^D) = 0, \qquad \mu = \int_{H^D} \mu_{x^D} d\mu^D(x^D).$$

The proof of the following result can be found in (Ambrosio et al., 2005, Corollary 9.4.6).

**Theorem 3** (KL divergence and orthogonal projection). *For every measures $\nu, \mu$ on $H$, we have*

$$\lim_{D \to \infty} \mathrm{KL}(\nu^D || \mu^D) = \mathrm{KL}(\nu || \mu).$$

## B.3 Lemmas

Before going through the proof of Theorem 1, we will need two lemmas. Similar results have been proved in (Sun et al., 2024; Balasubramanian et al., 2022; Vempala and Wibisono, 2019) for the finite-dimensional setting.

**Lemma 1.** *Let Assumptions 1 and 4 hold. Consider the stochastic process defined by*

$$X_t := X_0 - tQ_0 + 2^{\frac{1}{2}} C^{\frac{\alpha}{2}} W_t, \quad with \quad Q_0 := Q_0(X_0), \ X_0 \sim \nu_0,$$

*where*

$$Q_0(X_0) = -C^{\alpha-1} S(\tau, X_0; \mu_0) - C^{\alpha} \nabla_{X_0} \log(\rho(\mathbf{y} - \mathcal{A}(X_0))),$$

*and $\{W_t\}_{t \geq 0}$ is a Wiener process on $H$ independent of $X_0$. Then, writing $\nu_t$ for the probability measure of $X_t$, we have*

$$\frac{d}{dt}\mathrm{KL}(\nu_t || \mu^{\mathbf{y}}) \leq -\frac{3}{4} \int \left\| C^{\frac{\alpha}{2}} \nabla_X \log \frac{d\nu_t}{d\mu^{\mathbf{y}}}(X) \right\|_H^2 d\nu_t(X)$$

$$+ \mathbb{E}_{\nu_t} \left[ \| C^{\frac{\alpha}{2}} \nabla_X \Phi_0(X) + C^{\frac{\alpha}{2}} C^{-1}_{\mu_0} X - C^{-\frac{\alpha}{2}} Q_0(X_0) \|_H^2 \right].$$

*Proof.* We extend the proof of Lemma 1 in Sun et al. (2024) to infinite dimensions. The main idea is to derive the evolution of the density of $\nu_t^D$, plug it into the the time derivative formula for $\mathrm{KL}(\nu_t^D || (\mu^{\mathbf{y}})^D)$, and then take the limit as $D \to \infty$.

**Step 1: Projecting the process onto a finite-dimensional subspace** Let $D > D_0$, where $D_0$ is defined in Assumption 4 as the number of modes contributing to the observations. Consider the stochastic process $X_t^D$ defined by

$$X_t^D := X_0^D - tQ_0^D + \sqrt{2P^D C^\alpha P^D}W_t, \quad X_0^D \sim \nu_0^D, \tag{12}$$

where

$$Q_0^D := -(C^{\alpha-1})^D S^D(\tau, X_0; \mu_0) - (C^\alpha)^D \nabla_{X_0^D} \log(\rho(\mathbf{y} - \mathcal{A}(X_0)))$$

$$= -\sum_{j=1}^D \lambda_j^{\alpha-1} S^{(j)}(\tau, X_0; \mu_0)e_j - \sum_{j=1}^D \lambda_j^\alpha \nabla_j \log(\rho(\mathbf{y} - \mathcal{A}(X_0)))e_j,$$

with

$$S^{(j)}(\tau, X_0; \mu_0) := \langle S(\tau, X_0; \mu_0), e_j \rangle, \qquad \nabla_j f := \langle \nabla f, e_j \rangle.$$

We observe that

$$X_t^D = P^D(X_t).$$

Since $X_t^D$ will stay in $H^D$ for all the times, we can view $X_t^D$ as a process on $\mathbb{R}^D$ and define the Lebesgue densities $\nu_t^D$ of $X_t^D$ there.

**Step 2: Deriving the evolution equation for $\nu_t^D$** For each $t > 0$, let $\nu_{t,0}^D$ denote the Lebesgue density of the joint distribution of $(X_t^D, X_0^D)$. Let $\nu_{t|0}^D$ be the density of the conditional distribution of $X_t^D$ conditioned on $X_0^D$, and $\nu_{0|t}^D$ be the density of the probability distribution of $X_0^D$ conditioned on $X_t^D$. We have the relation

$$\nu_{t,0}^D(X^D, X_0^D) = \nu_{t|0}^D(X^D|X_0^D)\nu_0^D(X_0^D) = \nu_{0|t}(X_0^D|X^D)\nu_t^D(X^D).$$

Since $S^D(\tau, X_0) = S^D(\tau, X_0^D)$ by Assumption 4, conditioning on $X_0^D$ we have that $Q_0^D$ is a constant vector. Then, the conditional distribution $\nu_{t|0}^D$ evolves according to the following Fokker-Planck equation:

$$\frac{\partial}{\partial t}\nu_{t|0}^D(X^D|X_0^D) = \text{div}_{X^D}\left(\nu_{t|0}^D(X^D|X_0^D)Q_0^D + (C^\alpha)^D\nabla_{X^D}\nu_{t|0}^D(X^D|X_0^D)\right).$$

To derive the evolution equation for the marginal distribution $\nu_t^D(X^D)$, we need to take the expectation over $X_0^D \sim \nu_0^D$. Multiplying both sides of the Fokker-Planck equation by $\nu_0^D(X_0^D)$ and integrating over $X_0^D$, we have

$$\frac{\partial}{\partial t}\nu_t^D(X^D)$$

$$= \int \left(\frac{\partial}{\partial t}\nu_{t|0}^D(X^D|X_0^D)\right)\nu_0^D(X_0^D)dX_0^D$$

$$= \int \text{div}_{X^D}\left(\nu_{t|0}^D(X^D|X_0^D)Q_0^D + (C^\alpha)^D\nabla_{X^D}\nu_{t|0}^D(X^D|X_0^D)\right)\nu_0^D(X_0^D)dX_0^D$$

$$= \int \text{div}_{X^D}\left(\nu_{t,0}^D(X^D, X_0^D)Q_0^D + (C^\alpha)^D\nabla_{X^D}\nu_{t,0}^D(X^D, X_0^D)\right)dX_0^D \tag{13}$$

$$= \text{div}_{X^D}\left(\nu_t^D(X^D)\int \nu_{0|t}^D(X_0^D|X^D)Q_0^D dX_0^D + (C^\alpha)^D\nabla_{X^D}\int \nu_{t,0}^D(X^D, X_0^D)dX_0^D\right)$$

$$= \text{div}_{X^D}\left(\nu_t^D(X^D)\mathbb{E}_{\nu_{0|t}^D}[Q_0^D|X_t^D = X^D] + (C^\alpha)^D\nabla_{X^D}\nu_t^D(X^D)\right).$$

**Step 3: Calculating the derivative of the KL divergence** The time derivative of $\text{KL}(\nu_t^D || (\mu^{\mathbf{y}})^D)$ is given by

$$\frac{d}{dt}\text{KL}(\nu_t^D || (\mu^{\mathbf{y}})^D)$$

$$= \frac{d}{dt}\int \nu_t^D(X^D)\log\frac{\nu_t^D(X^D)}{(\mu^{\mathbf{y}})^D(X^D)}dX^D$$

$$= \int \frac{\partial \nu_t^D}{\partial t}\log\frac{\nu_t^D(X^D)}{(\mu^{\mathbf{y}})^D(X^D)}dX^D + \int \nu_t^D(X^D)\frac{\partial}{\partial t}\log\frac{\nu_t^D(X^D)}{(\mu^{\mathbf{y}})^D(X^D)}dX^D$$

$$= \int \frac{\partial \nu_t^D}{\partial t}\log\frac{\nu_t^D(X^D)}{(\mu^{\mathbf{y}})^D(X^D)}dX^D + \int \nu_t^D(X^D)\frac{(\mu^{\mathbf{y}})^D(X^D)}{\nu_t^D(X^D)}\frac{1}{(\mu^{\mathbf{y}})^D(X^D)}\frac{\partial \nu_t^D(X^D)}{\partial t}dX^D$$

$$= \int \frac{\partial \nu_t^D}{\partial t}\log\frac{\nu_t^D(X^D)}{(\mu^{\mathbf{y}})^D(X^D)}dX^D + \frac{\partial}{\partial t}\int \nu_t^D(X^D)dX^D$$

$$= \int \frac{\partial \nu_t^D}{\partial t}\log\frac{\nu_t^D(X^D)}{(\mu^{\mathbf{y}})^D(X^D)}dX^D.$$

By using the evolution equation for $\nu_t^D$ found in (13), we can derive

$$\frac{d}{dt}\text{KL}(\nu_t^D || (\mu^{\mathbf{y}})^D)$$

$$= \int \frac{\partial \nu_t^D(X^D)}{\partial t}\log\frac{\nu_t^D(X^D)}{(\mu^{\mathbf{y}})^D(X^D)}dX^D$$

$$= \int \text{div}_{X^D}\left(\left(\nu_t^D(X^D)\mathbb{E}_{\nu_{0|t}^D}[Q_0^D|X_t^D = X^D] + (C^\alpha)^D\nabla_{X^D}\nu_t^D(X^D)\right)\right)\log\frac{\nu_t^D(X^D)}{(\mu^{\mathbf{y}})^D(X^D)}dX^D$$

$$= -\int \left\langle \nu_t^D(X^D)\mathbb{E}_{\nu_{0|t}^D}[Q_0^D|X_t^D = X^D] + (C^\alpha)^D\nabla_{X^D}\nu_t^D(X^D), \nabla_{X^D}\log\frac{\nu_t^D(X^D)}{(\mu^{\mathbf{y}})^D(X^D)}\right\rangle dX^D$$

$$= -\int \left\langle \nu_t^D(X^D)\left(\mathbb{E}_{\nu_{0|t}^D}[Q_0^D|X_t^D = X^D] + (C^\alpha)^D\nabla_{X^D}\log\frac{\nu_t^D(X^D)}{(\mu^{\mathbf{y}})^D(X^D)}\right.\right.$$

$$\left.\left.+(C^\alpha)^D\nabla_{X^D}\log(\mu^{\mathbf{y}})^D(X^D)\right), \nabla_{X^D}\log\frac{\nu_t^D(X^D)}{(\mu^{\mathbf{y}})^D(X^D)}\right\rangle dX^D$$

$$= -\int \left\langle (C^\alpha)^D\nabla_{X^D}\log\frac{\nu_t^D(X^D)}{(\mu^{\mathbf{y}})^D(X^D)}, \nabla_{X^D}\log\frac{\nu_t^D(X^D)}{(\mu^{\mathbf{y}})^D(X^D)}\right\rangle \nu_t^D(X^D)dX^D$$

$$-\int \left\langle (C^\alpha)^D\nabla_{X^D}\log(\mu^{\mathbf{y}})^D(X^D) + \mathbb{E}_{\nu_{0|t}^D}[Q_0^D|X_t^D = X^D],\right.$$

$$\left.\nabla_{X^D}\log\frac{\nu_t^D(X^D)}{(\mu^{\mathbf{y}})^D(X^D)}\right\rangle \nu_t^D(X^D)dX^D.$$

**Step 4: Factorising $\nu_t$ and $\mu^{\mathbf{y}}$ into product of marginals** As a consequence of Assumption 4 and the definition of $X_t$, $\nu_t$ can be factorised into two blocks $(1 : D_0)$ and $(D_0 + 1 : \infty)$. The latter can be further factorised into a product of marginals, since $S$ can be diagonalised and the likelihood does not depend on $P^{D_0+1:\infty}(X_0)$, once more by Assumption 4. More precisely, we have

$$\nu_t(X) = \nu_t^{D_0}(X^{D_0})\prod_{j=D_0+1}^{\infty}\nu_t^{(j)}(X^{(j)}),$$

where $\nu_t^{D_0}$ is equivalent to $(\mu^{\mathbf{y}})^{D_0}$ (they both have densities with respect to the Lebesgue measure over $\mathbb{R}^{D_0}$) and each $\nu_t^{(j)}$ is equivalent to $(\mu^{\mathbf{y}})^{(j)}$. Then we have that

$$\frac{d\nu_t}{d\mu^{\mathbf{y}}}(X) = \left(\frac{\nu_t^{D_0}}{(\mu^{\mathbf{y}})^{D_0}}(X^{D_0})\right)\prod_{j=D_0+1}^{\infty}\frac{\nu_t^{(j)}}{(\mu^{\mathbf{y}})^{(j)}}(X^{(j)}).$$

In particular, for any $D > D_0$,

$$\frac{d\nu_t}{d\mu^{\mathbf{y}}}(X) = \left(\frac{\nu_t^D}{(\mu^{\mathbf{y}})^D}(X^D)\right)\prod_{j=D+1}^{\infty}\frac{\nu_t^{(j)}}{(\mu^{\mathbf{y}})^{(j)}}(X^{(j)}),$$

hence

$$C^{\frac{\alpha}{2}}\nabla_X \log \frac{d\nu_t}{d\mu^{\mathbf{y}}}(X)$$

$$= (C^{\frac{\alpha}{2}})^D \nabla_{X^D} \log\left(\frac{d\nu_t^D}{d(\mu^{\mathbf{y}})^D}(X^D)\right) + (C^{\frac{\alpha}{2}})^{D+1:\infty}\nabla_{X^{D+1:\infty}}\log\left(\prod_{j=D+1}^{\infty}\frac{\nu_t^{(j)}}{(\mu^{\mathbf{y}})^{(j)}}(X^{(j)})\right). \tag{14}$$

**Step 5: Taking the limit as $D \to \infty$**  Assume that

$$\int_H \left\|C^{\frac{\alpha}{2}}\nabla_X \log \frac{d\nu_t}{d\mu^{\mathbf{y}}}(X)\right\|_H^2 d\nu_t(X) < +\infty. \tag{15}$$

By Theorem 2, disintegrating $\nu_t$ with respect to $X^D$ yields

$$\int_H \left\|C^{\frac{\alpha}{2}}\nabla_X \log \frac{d\nu_t}{d\mu^{\mathbf{y}}}(X)\right\|_H^2 d\nu_t(X)$$

$$= \int_{H^D}\int_{H^{D+1:\infty}} \left\|C^{\frac{\alpha}{2}}\nabla_X \log \frac{d\nu_t}{d\mu^{\mathbf{y}}}(X)\right\|_{H^D}^2 d(\nu_t)_{X^D}(X^{D+1:\infty})d\nu_t^D(X^D).$$

We get

$$\int_H \left\|C^{\frac{\alpha}{2}}\nabla_X \log \frac{d\nu_t}{d\mu^{\mathbf{y}}}(X)\right\|_H^2 d\nu_t(X)$$

$$= \int_{H^D}\int_{H^{D+1:\infty}} \left\|C^{\frac{\alpha}{2}}\nabla_X \log \frac{d\nu_t}{d\mu^{\mathbf{y}}}(X)\right\|_H^2 d(\nu_t)_{X^D}(X^{D+1:\infty})d\nu_t^D(X^D)$$

$$= \int_{H^D}\int_{H^{D+1:\infty}} \left\|(C^{\frac{\alpha}{2}})^D \nabla_{X^D}\log\left(\frac{\nu_t^D}{(\mu^{\mathbf{y}})^D}(X^D)\right)\right.$$

$$\left. +(C^{\frac{\alpha}{2}})^{D+1:\infty}\nabla_{X^{D+1:\infty}}\log\left(\prod_{j=D+1}^{\infty}\frac{\nu_t^{(j)}}{(\mu^{\mathbf{y}})^{(j)}}(X^{(j)})\right)\right\|_H^2 d(\nu_t)_{X^D}(X^{D+1:\infty})d\nu_t^D(X^D)$$

$$= \int_{H^D} \left\|(C^{\frac{\alpha}{2}})^D \nabla_{X^D}\log\left(\frac{\nu_t^D}{(\mu^{\mathbf{y}})^D}(X^D)\right)\right\|_{H^D}^2 d\nu_t^D(X^D)$$

$$+ \int_{H^D}\int_{H^{D+1:\infty}} \sum_{j=D+1}^{\infty} \lambda_j^{\alpha}\left|\nabla_j \log\left(\frac{\nu_t^{(j)}}{(\mu^{\mathbf{y}})^{(j)}}(X^{(j)})\right)\right|^2 d(\nu_t)_{X^D}(X^{D+1:\infty})d\nu_t^D(X^D).$$

By (15), it follows that

$$\int_{H^D}\int_{H^{D+1:\infty}} \sum_{j=D+1}^{\infty} \lambda_j^{\alpha}\left|\nabla_j \log\left(\frac{\nu_t^{(j)}}{(\mu^{\mathbf{y}})^{(j)}}(X^{(j)})\right)\right|^2 d(\nu_t)_{X^D}(X^{D+1:\infty})\nu_t^D(X^D)dX^D \overset{D\to+\infty}{\longrightarrow} 0.$$

This means that

$$\lim_{D\to\infty}\int_{H^D}\left\|(C^{\frac{\alpha}{2}})^D \nabla_{X^D}\log\frac{d\nu_t^D}{d(\mu^{\mathbf{y}})^D}(X^D)\right\|_H^2 d\nu_t^D(X^D) = \int_H \left\|C^{\frac{\alpha}{2}}\nabla_X \log \frac{d\nu_t}{d\mu^{\mathbf{y}}}(X)\right\|_H^2 d\nu_t(X). \tag{16}$$

We now take the limit in

$$\frac{d}{dt}\mathrm{KL}(\nu_t^D\|(\mu^{\mathbf{y}})^D)$$

$$= -\int \left\langle (C^{\alpha})^D\nabla_{X^D}\log\frac{\nu_t^D(X^D)}{(\mu^{\mathbf{y}})^D(X^D)}, \nabla_{X^D}\log\frac{\nu_t^D(X^D)}{(\mu^{\mathbf{y}})^D(X^D)}\right\rangle \nu_t^D(X^D)dX^D$$

$$\quad -\int \left\langle (C^{\alpha})^D\nabla_{X^D}\log(\mu^{\mathbf{y}})^D(X^D)+\mathbb{E}_{\nu_{0|t}^D}[Q_0^D|X_t^D=X^D], \nabla_{X^D}\log\frac{\nu_t^D(X^D)}{(\mu^{\mathbf{y}})^D(X^D)}\right\rangle d\nu_t^D(X^D)$$

$$= -\int \left\langle (C^{\alpha})^D\nabla_{X^D}\log\frac{\nu_t^D(X^D)}{(\mu^{\mathbf{y}})^D(X^D)}, \nabla_{X^D}\log\frac{\nu_t^D(X^D)}{(\mu^{\mathbf{y}})^D(X^D)}\right\rangle \nu_t^D(X^D)dX^D$$

$$\quad -\int \left\langle (C^{\frac{\alpha}{2}})^D\nabla_{X^D}\log(\mu^{\mathbf{y}})^D(X^D) + (C^{-\frac{\alpha}{2}})^D\mathbb{E}_{\nu_{0|t}^D}[Q_0^D|X_t^D=X^D],\right.$$

$$\qquad \left. (C^{\frac{\alpha}{2}})^D\nabla_{X^D}\log\frac{\nu_t^D(X^D)}{(\mu^{\mathbf{y}})^D(X^D)}\right\rangle d\nu_t^D(X^D)dX^D.$$

By Theorem 3, we get

$$\frac{d}{dt}\mathrm{KL}(\nu_t^D\|(\mu^{\mathbf{y}})^D) \to \frac{d}{dt}\mathrm{KL}(\nu_t\|\mu^{\mathbf{y}})$$

for $D\to\infty$. By Eq. (16), we have

$$\lim_{D\to\infty} -\int \left\langle (C^{\alpha})^D\nabla_{X^D}\log\frac{\nu_t^D(X^D)}{(\mu^{\mathbf{y}})^D(X^D)}, \nabla_{X^D}\log\frac{\nu_t^D(X^D)}{(\mu^{\mathbf{y}})^D(X^D)}\right\rangle \nu_t^D(X^D)dX^D$$

$$= -\int \left\|C^{\frac{\alpha}{2}}\nabla_X\log\frac{d\nu_t}{d\mu^{\mathbf{y}}}(X)\right\|_H^2 d\nu_t(X).$$

We apply Young's inequality

$$-\int \left\langle (C^{\frac{\alpha}{2}})^D\nabla_{X^D}\log(\mu^{\mathbf{y}})^D(X^D) + (C^{-\frac{\alpha}{2}})^D\mathbb{E}_{\nu_{0|t}^D}[Q_0^D|X_t^D=X^D],\right.$$

$$\qquad \left. (C^{\frac{\alpha}{2}})^D\nabla_{X^D}\log\frac{\nu_t^D(X^D)}{(\mu^{\mathbf{y}})^D(X^D)}\right\rangle \nu_t^D(X^D)dX^D$$

$$\leq \frac{1}{4}\int \left\langle (C^{\alpha})^D\nabla_{X^D}\log\frac{\nu_t^D(X^D)}{(\mu^{\mathbf{y}})^D(X^D)}, \nabla_{X^D}\log\frac{\nu_t^D(X^D)}{(\mu^{\mathbf{y}})^D(X^D)}\right\rangle \nu_t^D(X^D)dX^D$$

$$\quad + \int \left\|-(C^{\frac{\alpha}{2}})^D\nabla_{X^D}\log(\mu^{\mathbf{y}})^D(X^D) - (C^{-\frac{\alpha}{2}})^DQ_0^D\right\|_H^2 \nu_t^D(X^D)dX^D.$$

We need to calculate

$$\lim_{D\to\infty}\left\{\int \left\|-(C^{\frac{\alpha}{2}})^D\nabla_{X^D}\log(\mu^{\mathbf{y}})^D(X^D) - (C^{-\frac{\alpha}{2}})^DQ_0^D\right\|_H^2 \nu_t^D(X^D)dX^D\right\},$$

as so far we have proved that

$$\frac{d}{dt}\mathrm{KL}(\nu_t\|\mu^{\mathbf{y}})$$

$$\leq -\frac{3}{4}\int \left\|C^{\frac{\alpha}{2}}\nabla_X\log\frac{d\nu_t(X)}{d\mu^{\mathbf{y}}(X)}\right\|_H^2 d\nu_t(X)$$

$$\quad + \lim_{D\to\infty}\left\{\int \left\|-(C^{\frac{\alpha}{2}})^D\nabla_{X^D}\log(\mu^{\mathbf{y}})^D(X^D) - (C^{-\frac{\alpha}{2}})^DQ_0^D\right\|_H^2 \nu_t^D(X^D)dX^D\right\}.$$

Recall that

$$Q_0^D = -(C^{\alpha-1})^DS^D(\tau, X_0^D;\mu_0) - \sum_{j=1}^{D}\lambda_j^{\alpha}\nabla_j\log(\rho(\mathbf{y}-\mathcal{A}(X_0)))e_j,$$

where $S^D(\tau, X_0; \mu_0) = S^D(\tau, X_0^D; \mu_0)$ follows from the separability assumptions on $\nu_0$. We have

$$\left\|-(C^{\frac{\alpha}{2}})^D \nabla_{X^D} \log(\mu^{\mathbf{y}})^D(X^D) - (C^{-\frac{\alpha}{2}})^D Q_0^D\right\|_H^2$$

$$= \left\|-(C^{\frac{\alpha}{2}})^D \nabla_{X^D} \log(\mu^{\mathbf{y}})^D(X^D) + (C^{\frac{\alpha}{2}-1})^D S^D(\tau, X_0^D; \mu_0) + \sum_{j=1}^D \lambda_j^{\frac{\alpha}{2}} \nabla_j \log(\rho(\mathbf{y} - \mathcal{A}(X_0))) e_j\right\|_H^2.$$

We would like to prove that

$$\lim_{D \to \infty} \left\{ \int \left\|-(C^{\frac{\alpha}{2}})^D \nabla_{X^D} \log(\mu^{\mathbf{y}})^D(X^D) + (C^{\frac{\alpha}{2}-1})^D S^D(\tau, X_0^D; \mu_0) \right. \right.$$

$$\left. \left. + \sum_{j=1}^D \lambda_j^{\frac{\alpha}{2}} \nabla_j \log(\rho(\mathbf{y} - \mathcal{A}(X_0))) e_j \right\|_H^2 \nu_t^D(X^D) dX^D \right\}$$

$$\leq \int \left\|-C^{\frac{\alpha}{2}} \nabla_X \Phi_0(X) - C^{\frac{\alpha}{2}} C_{\mu_0}^{-1} X + C^{\frac{\alpha}{2}-1} S(\tau, X_0; \mu_0) \right.$$

$$\left. + C^{\frac{\alpha}{2}} \nabla_{X_0} \log(\rho(\mathbf{y} - \mathcal{A}(X_0)))\right\|_H^2 d\nu_t(X).$$

First, notice that

$$(C^{\frac{\alpha}{2}})^D \nabla_{X^D} \log(\mu^{\mathbf{y}})^D(X^D)$$

$$= (C^{\frac{\alpha}{2}})^D \nabla_{X^D} \log \frac{(\mu^{\mathbf{y}})^D(X^D)}{\mathcal{N}(0, C_{\mu_0}^D)(X^D)} + (C^{\frac{\alpha}{2}})^D \nabla_{X^D} \log \mathcal{N}(0, C_{\mu_0}^D)(X^D).$$

Then, since

$$(C^{\frac{\alpha}{2}})^D \nabla_{X^D} \log \mathcal{N}(0, C_{\mu_0}^D)(X^D) = -(C^{\frac{\alpha}{2}})^D (C_{\mu_0}^D)^{-1} X^D,$$

we get

$$\left\|-(C^{\frac{\alpha}{2}})^D \nabla_{X^D} \log(\mu^{\mathbf{y}})^D(X^D) - (C^{-\frac{\alpha}{2}})^D Q_0^D\right\|_H^2$$

$$= \left\|-(C^{\frac{\alpha}{2}})^D \nabla_{X^D} \log \frac{(\mu^{\mathbf{y}})^D(X^D)}{\mathcal{N}(0, C_{\mu_0}^D)(X^D)} + (C^{\frac{\alpha}{2}})^D (C_{\mu_0}^D)^{-1} X^D + (C^{\frac{\alpha}{2}-1})^D S^D(\tau, X_0^D; \mu_0) \right.$$

$$\left. + \sum_{j=1}^D \lambda_j^{\frac{\alpha}{2}} \nabla_j \log(\rho(\mathbf{y} - \mathcal{A}(X_0))) e_j\right\|_H^2.$$

By Assumption 4 and the fact that the likelihood does not depend on $P^{D+1:\infty}(X_0)$ for any $D > D_0$, and since $d\mathcal{N}(0, C_{\mu_0}) = d\mathcal{N}(0, C_{\mu_0}^D) d\mathcal{N}(0, C_{\mu_0}^{D+1:\infty})$ (see (Da Prato, 2006, Definition 1.5.2)), we can follow the same procedure that led to (16) and prove that

$$\lim_{D \to \infty} \left\{ \int_{H^D} \left\|(C^{\frac{\alpha}{2}})^D \nabla_{X^D} \log(\mu^{\mathbf{y}})^D(X^D) - (C^{-\frac{\alpha}{2}})^D Q_0^D\right\|_H^2 \nu_t^D(X^D) dX^D \right\}$$

$$= \int_H \left\|C^{\frac{\alpha}{2}} \nabla_X \Phi_0(X) + C^{\frac{\alpha}{2}} C_{\mu_0}^{-1} X + C^{\frac{\alpha}{2}-1} S(\tau, X_0; \mu_0) \right.$$

$$\left. + C^{\frac{\alpha}{2}} \nabla_{X_0} \log(\rho(\mathbf{y} - \mathcal{A}(X_0)))\right\|_H^2 d\nu_t(X),$$

where we used $d\mu^{\mathbf{y}} \propto \exp(-\Phi_0) d\mathcal{N}(0, C_{\mu_0})$ as per (2). Putting everything together, we have

$$\frac{d}{dt} \mathrm{KL}(\nu_t || \mu^{\mathbf{y}}) \leq -\frac{3}{4} \int \left\|C^{\frac{\alpha}{2}} \nabla_X \log \frac{d\nu_t}{d\mu^{\mathbf{y}}}(X)\right\|_H^2 d\nu_t(X)$$

$$+ \mathbb{E}_{\nu_t}\left[\|C^{\frac{\alpha}{2}} \nabla_X \Phi_0(X) + C^{\frac{\alpha}{2}} C_{\mu_0}^{-1} X - C^{-\frac{\alpha}{2}} Q_0(X_0)\|_H^2\right],$$

which ends the proof of the lemma. □

**Lemma 2.** *Define* $\mathcal{G} : H \to H$ *as*

$$\mathcal{G}(X) := -C^{\alpha-1} S_\theta(\tau, X; \mu_0) - C^\alpha \nabla_X \log(\rho(\mathbf{y} - \mathcal{A}(X))), \tag{17}$$

*where $S_\theta$ represents a neural network approximating the score defined in (5). Let Assumptions 1, 2, and 4 hold. It holds that*

$$\mathbb{E}_{\nu_t}[\|C^{-\frac{\alpha}{2}}\mathcal{G}(X)\|_H^2] \leq 2\int \left\|C^{\frac{\alpha}{2}}\nabla_X \log\frac{d\nu_t}{d\mu^{\mathbf{y}}}(X)\right\|_H^2 d\nu_t(X) + 4\mathrm{Tr}(C^\alpha)L_{\Phi_0}$$
$$+ 2\mathbb{E}_{\nu_t}[\|C^{\frac{\alpha}{2}}\nabla\Phi_0(X) + C^{\frac{\alpha}{2}}C_{\mu_0}^{-1}X - C^{-\frac{\alpha}{2}}\mathcal{G}(X)\|_H^2].$$

*Proof.* First, notice that

$$\mathcal{G}(X) = \mathcal{G}(X) - (C^\alpha\nabla_X\Phi_0(X) + C^\alpha C_{\mu_0}^{-1}X) + C^\alpha\nabla_X\Phi_0(X) + C^\alpha C_{\mu_0}^{-1}X.$$

We apply Young's inequality to get

$$\mathbb{E}_{\nu_t}[\|C^{-\frac{\alpha}{2}}\mathcal{G}(X)\|_H^2]$$
$$\leq 2\mathbb{E}_{\nu_t}[\|C^{\frac{\alpha}{2}}\nabla_X\Phi_0(X) + C^{\frac{\alpha}{2}}C_{\mu_0}^{-1}X\|_H^2]$$
$$+ 2\mathbb{E}_{\nu_t}[\|C^{\frac{\alpha}{2}}\nabla_X\Phi_0(X) + C^{\frac{\alpha}{2}}C_{\mu_0}^{-1}X - C^{-\frac{\alpha}{2}}\mathcal{G}(X)\|_H^2].$$

We study the first term of the inequality above. Notice that

$$\mathbb{E}_{\nu_t}[\|C^{\frac{\alpha}{2}}\nabla_X\Phi_0(X) + C^{\frac{\alpha}{2}}C_{\mu_0}^{-1}X\|_H^2] = \mathbb{E}_{\nu_t}\left[\left\|-C^{\frac{\alpha}{2}}\nabla_X\log\left(\frac{d\mu^{\mathbf{y}}}{d\mu_0}\right)(X) + C^{\frac{\alpha}{2}}C_{\mu_0}^{-1}X\right\|_H^2\right],$$

where we used the relation

$$\nabla_X\Phi_0(X) = -\nabla_X\log\left(\frac{d\mu^{\mathbf{y}}}{d\mu_0}\right)(X).$$

With the same arguments as in the proof of the previous lemma, we can write

$$\mathbb{E}_{\nu_t}\left[\left\|-C^{\frac{\alpha}{2}}\nabla_X\log\left(\frac{d\mu^{\mathbf{y}}}{d\mu_0}\right)(X) + C^{\frac{\alpha}{2}}C_{\mu_0}^{-1}X\right\|_H^2\right]$$
$$= \lim_{D\to\infty}\mathbb{E}_{\nu_t^D}\left[\left\|-(C^{\frac{\alpha}{2}})^D\nabla_{X^D}\log\frac{(\mu^{\mathbf{y}})^D(X^D)}{\mu_0^D(X^D)} + (C^{\frac{\alpha}{2}})^D(C_{\mu_0}^D)^{-1}X^D\right\|_{H^D}^2\right]$$
$$= \lim_{D\to\infty}\mathbb{E}_{\nu_t^D}\left[\left\|-(C^{\frac{\alpha}{2}})^D\nabla_{X^D}\log(\mu^{\mathbf{y}})^D(X^D) + (C^{\frac{\alpha}{2}})^D\nabla_{X^D}\log\mu_0^D(X^D)\right.\right.$$
$$\left.\left.+(C^{\frac{\alpha}{2}})^D(C_{\mu_0}^D)^{-1}X^D\right\|_{H^D}^2\right]$$

Since $\mu_0 = \mathcal{N}(0, C_{\mu_0})$, we have

$$(C^{\frac{\alpha}{2}})^D\nabla_{X^D}\log\mu_0^D(X^D) = -(C^{\frac{\alpha}{2}})^D(C_{\mu_0}^D)^{-1}X^D.$$

It follows that

$$\mathbb{E}_{\nu_t}\left[\left\|-C^{\frac{\alpha}{2}}\nabla_X\log\left(\frac{d\mu^{\mathbf{y}}}{d\mu_0}\right)(X) + C^{\frac{\alpha}{2}}C_{\mu_0}^{-1}X\right\|_H^2\right]$$
$$= \lim_{D\to\infty}\mathbb{E}_{\nu^D}\left[\|-(C^{\frac{\alpha}{2}})^D\nabla_{X^D}\log(\mu^{\mathbf{y}})^D(X^D)\|_{H^D}^2\right].$$

We can derive

$$\mathbb{E}_{\nu_t^D}\left[\left\|-(C^{\frac{\alpha}{2}})^D\nabla_{X^D}\log((\mu^{\mathbf{y}})^D)(X^D)\right\|_{H^D}^2\right]$$

$$= \mathbb{E}_{\nu_t^D}\left[\left\|-(C^{\frac{\alpha}{2}})^D\nabla_{X^D}\log(\mu^{\mathbf{y}})^D(X^D)+(C^{\frac{\alpha}{2}})^D\nabla\log\nu_t^D(X^D)\right.\right.$$
$$\left.\left.-(C^{\frac{\alpha}{2}})^D\nabla\log\nu_t^D(X^D)\right\|_{H^D}^2\right]$$

$$= \mathbb{E}_{\nu_t^D}\left[\left\|-(C^{\frac{\alpha}{2}})^D\nabla_{X^D}\log(\mu^{\mathbf{y}})^D(X^D)+(C^{\frac{\alpha}{2}})^D\nabla\log\nu_t^D(X^D)\right\|_{H^D}^2\right.$$
$$+2\langle(C^{\frac{\alpha}{2}})^D\nabla_{X^D}\log(\mu^{\mathbf{y}})^D(X^D)-(C^{\frac{\alpha}{2}})^D\nabla_{X^D}\log\nu_t^D(X^D),$$
$$\left.(C^{\frac{\alpha}{2}})^D\nabla_{X^D}\log\nu_t^D(X^D)\rangle+\|(C^{\frac{\alpha}{2}})^D\nabla_{X^D}\log\nu_t^D(X^D)\|_{H^D}^2\right]$$

$$= \mathbb{E}_{\nu_t^D}\left[\left\|-(C^{\frac{\alpha}{2}})^D\nabla_{X^D}\log(\mu^{\mathbf{y}})^D(X^D)+(C^{\frac{\alpha}{2}})^D\nabla\log\nu_t^D(X^D)\right\|_{H^D}^2\right.$$
$$+\langle 2(C^{\frac{\alpha}{2}})^D\nabla_{X^D}\log(\mu^{\mathbf{y}})^D(X^D)-(C^{\frac{\alpha}{2}})^D\nabla_{X^D}\log\nu_t^D(X^D),$$
$$\left.(C^{\frac{\alpha}{2}})^D\nabla_{X^D}\log\nu_t^D(X^D)\rangle\right]$$

$$\leq \int_{H^D}\left\|(C^{\frac{\alpha}{2}})^D\nabla_{X^D}\log\frac{\nu_t^D(X^D)}{(\mu^{\mathbf{y}})^D(X^D)}\right\|_{H^D}^2\nu_t^D(X^D)dX^D$$
$$+2\mathbb{E}_{\nu_t^D}[\langle(C^{\frac{\alpha}{2}})^D\nabla_{X^D}\log(\mu^{\mathbf{y}})^D(X^D),(C^{\frac{\alpha}{2}})^D\nabla_{X^D}\log\nu_t^D(X^D)\rangle],$$

where in the last inequality we used the fact that

$$-\mathbb{E}_{\nu^D}[\|(C^{\frac{\alpha}{2}})^D\nabla_{X^D}\log\nu_t^D(X^D)\|_H^2]\leq 0.$$

For Lemma 1, we proved

$$\lim_{D\to\infty}\int_{H^D}\left\|(C^{\frac{\alpha}{2}})^D\nabla_{X^D}\log\frac{\nu_t^D(X^D)}{(\mu^{\mathbf{y}})^D(X^D)}\right\|_{H^D}^2\nu_t^D(X^D)dX^D$$
$$=\int_H\left\|C^{\frac{\alpha}{2}}\nabla_X\log\frac{d\nu_t}{d\mu^{\mathbf{y}}}(X)\right\|_H^2 d\nu_t(X).$$

We notice that

$$\mathbb{E}_{\nu_t^D}[\langle(C^{\frac{\alpha}{2}})^D\nabla_{X^D}\log(\mu^{\mathbf{y}})^D(X^D),(C^{\frac{\alpha}{2}})^D\nabla_{X^D}\log\nu_t^D(X^D)\rangle]$$

$$=\int_{H^D}\langle(C^{\frac{\alpha}{2}})^D\nabla_{X^D}\log(\mu^{\mathbf{y}})^D(X^D),(C^{\frac{\alpha}{2}})^D\nabla_{X^D}\log\nu_t^D(X^D)\rangle\nu_t^D(X^D)dX^D$$

$$=\int_{H^D}\langle(C^{\frac{\alpha}{2}})^D\nabla_{X^D}\log(\mu^{\mathbf{y}})^D(X^D),(C^{\frac{\alpha}{2}})^D\nabla_{X^D}\nu_t^D(X^D)\rangle dX^D$$

$$=\int_{H^D}\langle(C^\alpha)^D\nabla_{X^D}\log(\mu^{\mathbf{y}})^D(X^D),\nabla_{X^D}\nu_t^D(X^D)\rangle dX^D$$

$$=-\int_{H^D}\mathrm{div}_{X^D}((C^\alpha)^D\nabla_{X^D}\log(\mu^{\mathbf{y}})^D(X^D))\nu_t^D(X^D)dX^D\leq\mathrm{Tr}(C^\alpha)L_{\Phi_0}.$$

In the second identity, we used $\nu_t^D(X^D)\nabla_{X^D}\log\nu_t^D(X^D)=\nabla_{X^D}\nu_t^D(X^D)$. In the fourth identity, we used

$$\mathrm{div}_{X^D}((C^\alpha)^D\nabla_{X^D}\log(\mu^{\mathbf{y}})^D(X^D)\nu_t^D(X^D))$$
$$=\mathrm{div}_{X^D}((C^\alpha)^D\nabla_{X^D}\log(\mu^{\mathbf{y}})^D(X^D))\nu_t^D(X^D)+\langle(C^\alpha)^D\nabla_{X^D}\log(\mu^{\mathbf{y}})^D(X^D),\nabla_{X^D}\nu_t^D(X^D)\rangle$$

and

$$\int_{H^D}\mathrm{div}_{X^D}\left((C^\alpha)^D\nabla_{X^D}\log(\mu^{\mathbf{y}})^D(X^D)\nu_t^D(X^D)\right)dX^D=0.$$

In the last inequality, we used

$$\nabla_{X^D}\log(\mu^{\mathbf{y}})^D(X^D)=\nabla_{X^D}\log\frac{(\mu^{\mathbf{y}})^D(X^D)}{\mu_0^D(X^D)}+\nabla_{X^D}\log\mu_0^D(X^D)$$
$$=-\nabla_{X^D}\tilde{\Phi}_0(X^D)+(C_{\mu_0}^D)^{-1}X^D,$$

with

$$\tilde{\Phi}_0(X^D) := \int_{H^{D+1:\infty}} \Phi_0(X^D, X^{D+1:\infty}) d(\mu^{\mathbf{y}})_{X^D}(X^{D+1:\infty}).$$

Then

$$-\int_{H^D} \mathrm{div}_{X^D}((C^\alpha)^D \nabla_{X^D} \log(\mu^{\mathbf{y}})^D)(X^D)\nu_t^D(X^D)dX^D$$

$$= \int_{H^D} \mathrm{div}_{X^D}((C^\alpha)^D \nabla_{X^D} \tilde{\Phi}_0)(X^D)\nu_t^D(X^D)dX^D - \sum_{j=1}^D \left(\frac{\lambda_j^\alpha}{\mu_{0j}}\right)\int_{H^D} \nu_t^D(X^D)dX^D$$

$$\leq \mathrm{Tr}(C^\alpha)L_{\Phi_0}\int_{H^D} \nu_t^D(X^D)dX^D - \sum_{j=1}^D \left(\frac{\lambda_j^\alpha}{\mu_{0j}}\right)$$

$$\leq \mathrm{Tr}(C^\alpha)L_{\Phi_0}.$$

This is because, since we assumed that $\nabla\Phi_0 \in C^1$ is Lipschitz continuous with a constant $L_{\Phi_0}$, so is $\nabla_{X^D}\tilde{\Phi}_0(X^D)$, and then

$$|\mathrm{div}_{X^D}((C^\alpha)^D \nabla_{X^D}\tilde{\Phi}_0(X^D))| \leq \mathrm{Tr}(C^\alpha)L_{\Phi_0}.$$

Combining all the inequalities above, we get

$$\mathbb{E}_{\nu_t}[\|C^{-\frac{\alpha}{2}}\mathcal{G}(X)\|_H^2] \leq 2\int_H \left\|C^{\frac{\alpha}{2}}\nabla_X \log\frac{d\nu_t}{d\mu^{\mathbf{y}}}(X)\right\|_H^2 d\nu_t(X) + 4\mathrm{Tr}(C^\alpha)L_{\Phi_0}$$

$$+ 2\mathbb{E}_{\nu_t}[\|C^{\frac{\alpha}{2}}\nabla\Phi_0(X) + C^{\frac{\alpha}{2}}C_{\mu_0}^{-1}X - C^{-\frac{\alpha}{2}}\mathcal{G}(X)\|_H^2].$$

The proof is complete. $\square$

### B.4 PROOF OF THEOREM 1

We are now ready to prove our convergence theorem.

*Proof of Theorem 1.* We construct the following interpolation for our method

$$X_t = X_{k\gamma} - (t - k\gamma)\left(-C^{\alpha-1}S_\theta(\tau, X_{k\gamma}; \mu_0) - C^\alpha \nabla_{X_{k\gamma}} \log(\rho(\mathbf{y} - \mathcal{A}(X_{k\gamma})))\right)$$

$$+ 2^{\frac{1}{2}}C^{\frac{\alpha}{2}}(W_t - W_{k\gamma}),$$

$$\text{for } t \in [k\gamma, (k+1)\gamma].$$

Let $\nu_t$ be the law of $X_t$. As in Lemma 2, define

$$\mathcal{G}(X, \tau) := -C^{\alpha-1}S_\theta(\tau, X; \mu_0) - C^\alpha \nabla_X \log(\rho(\mathbf{y} - \mathcal{A}(X))).$$

As a consequence of Corollary 1, the distance of $C^{-\frac{\alpha}{2}}\mathcal{G}(X, \tau)$ from $C^{\frac{\alpha}{2}}\nabla_X\Phi_0(X) + C^{\frac{\alpha}{2}}C_{\mu_0}^{-1}X$ is given by

$$\|C^{\frac{\alpha}{2}}\nabla\Phi_0(X) + C^{\frac{\alpha}{2}}C_{\mu_0}^{-1}X - C^{-\frac{\alpha}{2}}\mathcal{G}(X, \tau)\|_H^2$$

$$\leq 2\|C^{\frac{\alpha}{2}}\nabla\Phi_0(X) + C^{\frac{\alpha}{2}}C_{\mu_0}^{-1}X + C^{\frac{\alpha}{2}-1}S(\tau, X; \mu_0) - C^{\frac{\alpha}{2}}\nabla\Phi_0(X)\|_H^2$$

$$+ 2\|C^{\frac{\alpha}{2}-1}(S_\theta(X, \tau; \mu_0) - S(X, \tau; \mu_0))\|_H^2$$

$$\text{(Plug-in Corollary 1)} \tag{18}$$

$$\leq 2\tau^2 K'^2\|X\|_H^2 + 2\|C^{\frac{\alpha}{2}-1}(S_\theta(X, \tau; \mu_0) - S(X, \tau; \mu_0))\|_H^2$$

$$\text{(Assumption 3)}$$

$$\leq 2\tau^2 K'^2\|X\|_H^2 + 2\|C\|^{\alpha-2}\epsilon_\tau^2,$$

where $K'^2 = \|C\|^{\alpha-2}(\|CC_{\mu_0}^{-1}\| + 1)^2\|CC_{\mu_0}^{-1}\|^2$ and $\|C\|$ is the spectral norm of $C$, i.e., its largest eigenvalue, and The last inequality is valid only if $\alpha \geq 2$. Note that, if we have a stronger control of the score approximation error, that is to say, if we assume that $\|C^{-\beta}(S_\theta(\tau, X; \mu_0) - S(\tau, X; \mu_0))\|_H \leq$

$\epsilon_{\beta,\tau}$ for some $\beta \geq 0$ instead of (11), then we can replace the upper bound $2\|C\|^{\alpha-2}\epsilon_\tau^2$ by $2\|C\|^{\alpha-2+2\beta}\epsilon_{\beta,\tau}^2$ and this new bound is valid for any $\alpha \geq 2 - 2\beta$.

From (17) we have

$$\mathbb{E}_{\nu_t}[\|C^{-\frac{\alpha}{2}}\mathcal{G}(X_t,\tau) - C^{-\frac{\alpha}{2}}\mathcal{G}(X_{k\gamma},\tau)\|_H^2]$$
$$\leq 2\mathbb{E}_{\nu_t}[\|C^{\frac{\alpha}{2}-1}S_\theta(X_t,\tau;\mu_0) - C^{\frac{\alpha}{2}-1}S_\theta(X_{k\gamma},\tau;\mu_0)\|_H^2]$$
$$+ 2\mathbb{E}_{\nu_t}[\|C^{\frac{\alpha}{2}}\nabla\Phi_0(X_t) - C^{\frac{\alpha}{2}}\nabla\Phi_0(X_{k\gamma})\|_H^2].$$

By Assumptions 2 and 3, we have

$$\mathbb{E}_{\nu_t}[\|C^{-\frac{\alpha}{2}}\mathcal{G}(X_t,\tau) - C^{-\frac{\alpha}{2}}\mathcal{G}(X_{k\gamma},\tau)\|_H^2] \leq 2(\|C\|^{\alpha-2}L_\tau^2 + \|C\|^\alpha L_{\Phi_0}^2)\mathbb{E}_{\nu_t}[\|X_t - X_{k\gamma}\|_H^2]$$
$$\leq 2L_\mathcal{G}^2\mathbb{E}_{\nu_t}[\|X_t - X_{k\gamma}\|_H^2], \tag{19}$$

where the last inequality is valid only if $\alpha \geq 2$ and

$$L_\mathcal{G} := \sqrt{\|C\|^{\alpha-2}L_\tau^2 + \|C\|^\alpha L_{\Phi_0}^2}.$$

From Lemma 1, we know for $t \in [k\gamma, (k+1)\gamma]$ that

$$\frac{d}{dt}\mathrm{KL}(\nu_t\|\mu^{\mathbf{y}}) \leq -\frac{3}{4}\int\left\|C^{\frac{\alpha}{2}}\nabla_{X_t}\log\frac{d\nu_t}{d\mu^{\mathbf{y}}}(X_t)\right\|_H^2 d\nu_t(X_t)$$
$$+ \mathbb{E}_{\nu_t}\left[\|C^{\frac{\alpha}{2}}\nabla\Phi_0(X_t) + C^{\frac{\alpha}{2}}C_{\mu_0}^{-1}X_t - C^{-\frac{\alpha}{2}}\mathcal{G}(X_{k\gamma},\tau)\|_H^2\right]. \tag{20}$$

The second term can be bounded via Young's inequality, (18) and (19):

$$\mathbb{E}_{\nu_t}\left[\|C^{\frac{\alpha}{2}}\nabla\Phi_0(X_t) + C^{\frac{\alpha}{2}}C_{\mu_0}^{-1}X_t - C^{-\frac{\alpha}{2}}\mathcal{G}(X_{k\gamma},\tau)\|_H^2\right]$$
$$\leq 2\mathbb{E}_{\nu_t}[\|C^{-\frac{\alpha}{2}}\mathcal{G}(X_t,\tau) - C^{-\frac{\alpha}{2}}\mathcal{G}(X_{k\gamma},\tau)\|_H^2]$$
$$+ 2\mathbb{E}_{\nu_t}[\|C^{\frac{\alpha}{2}}\nabla\Phi_0(X_t) + C^{\frac{\alpha}{2}}C_{\mu_0}^{-1}X_t - C^{-\frac{\alpha}{2}}\mathcal{G}(X_t,\tau)\|_H^2]$$
$$\leq 4L_\mathcal{G}^2\mathbb{E}_{\nu_t}[\|X_t - X_{k\gamma}\|_H^2] + 4(\tau^2 K^2 + \|C\|^{\alpha-2}\epsilon_\tau^2), \tag{21}$$

where $K^2 = K'^2\sup_{t\in[0,N\gamma]}\mathbb{E}[\|X_t\|_H^2]$. We can bound the first term of the inequality above via

$$\mathbb{E}_{\nu_t}[\|X_t - X_{k\gamma}\|_H^2]$$
$$\leq 2(t-k\gamma)^2\|C\|^\alpha\mathbb{E}_{\nu_t}\left[\left\|-C^{\frac{\alpha}{2}-1}S_\theta(\tau,X_{k\gamma};\mu_0) - C^{\frac{\alpha}{2}}\nabla_{X_{k\gamma}}\log(\rho(\mathbf{y}-\mathcal{A}(X_{k\gamma})))\right\|_H^2\right]$$
$$+ 4\mathbb{E}_{\nu_t}[\|C^{\frac{\alpha}{2}}(W_t - W_{k\gamma})\|_H^2]$$
$$\leq 2(t-k\gamma)^2\|C\|^\alpha\left(2\mathbb{E}_{\nu_t}\left[\left\|-C^{\frac{\alpha}{2}-1}S_\theta(\tau,X_t;\mu_0) - C^{\frac{\alpha}{2}}\nabla_{X_t}\log(\rho(\mathbf{y}-\mathcal{A}(X_t)))\right\|_H^2\right]\right.$$
$$\left.+ 4L_\mathcal{G}^2\mathbb{E}_{\nu_t}[\|X_{k\gamma} - X_t\|_H^2]\right) + 4\mathrm{Tr}(C^\alpha)(t-k\gamma),$$

where for the last step we used Young's inequality and (19). Rearranging the terms yields

$$(1 - 8(t-k\gamma)^2\|C\|^\alpha L_\mathcal{G}^2)\mathbb{E}_{\nu_t}[\|X_{k\gamma} - X_t\|_H^2]$$
$$\leq 4(t-k\gamma)^2\|C\|^\alpha\mathbb{E}_{\nu_t}\left[\left\|-C^{\frac{\alpha}{2}-1}S_\theta(\tau,X_t;\mu_0) - C^{\frac{\alpha}{2}}\nabla_{X_t}\log(\rho(\mathbf{y}-\mathcal{A}(X_t)))\right\|_H^2\right]$$
$$+ 4\mathrm{Tr}(C^\alpha)(t-k\gamma),$$

which can be simplified by letting $\gamma \leq \frac{1}{4\sqrt{\|C\|^\alpha L_\mathcal{G}}} \Rightarrow 1 - 8(t-k\gamma)^2\|C\|^\alpha L_\mathcal{G}^2 \geq 1 - 8\gamma^2\|C\|^\alpha L_\mathcal{G}^2 \geq \frac{1}{2}$. Therefore, when $\gamma \leq \frac{1}{4\sqrt{\|C\|^\alpha L_\mathcal{G}}}$, it holds that

$$\mathbb{E}_{\nu_t}[\|X_{k\gamma} - X_t\|_H^2]$$
$$\leq 8(t-k\gamma)^2\|C\|^\alpha\mathbb{E}_{\nu_t}\left[\left\|-C^{\frac{\alpha}{2}-1}S_\theta(\tau,X_t;\mu_0) - C^{\frac{\alpha}{2}}\nabla_{X_t}\log(\rho(\mathbf{y}-\mathcal{A}(X_t)))\right\|_H^2\right] \tag{22}$$
$$+ 8\mathrm{Tr}(C^\alpha)(t-k\gamma).$$

By plugging (22) and (21) into (20) and invoking Lemma 2, we can obtain

$$\frac{d}{dt}\text{KL}(\nu_t||\mu^{\mathbf{y}})$$

(Plug-in Eq. (20) and Eq. (21))

$$\leq -\frac{3}{4}\int\left\|C^{\frac{\alpha}{2}}\nabla_{X_t}\log\frac{d\nu_t}{d\mu^{\mathbf{y}}}(X_t)\right\|_H^2 d\nu_t(X_t) + 4L_{\mathcal{G}}^2\mathbb{E}_{\nu_t}[\|X_t - X_{k\gamma}\|_H^2] + 4(\tau^2 K^2 + \|C\|^{\alpha-2}\epsilon_\tau^2)$$

(Plug-in Eq. (22))

$$\leq -\frac{3}{4}\int\left\|C^{\frac{\alpha}{2}}\nabla_{X_t}\log\frac{d\nu_t}{d\mu^{\mathbf{y}}}(X_t)\right\|_H^2 d\nu_t(X_t)$$

$$+ 32(t-k\gamma)^2\|C\|^\alpha L_{\mathcal{G}}^2\mathbb{E}_{\nu_t}\left[\left\|-C^{\frac{\alpha}{2}-1}S_\theta(\tau, X_t; \mu_0) - C^{\frac{\alpha}{2}}\nabla_{X_t}\log(\rho(\mathbf{y} - \mathcal{A}(X_t)))\right\|_H^2\right]$$

$$+ 32\text{Tr}(C^\alpha)(t-k\gamma)L_{\mathcal{G}}^2 + 4(\tau^2 K^2 + \|C\|^{\alpha-2}\epsilon_\tau^2)$$

(Plug-in Lemma 2 with $\|C^{\frac{\alpha}{2}}(\nabla\Phi_0(X) + C_{\mu_0}^{-1}X) - C^{-\frac{\alpha}{2}}\mathcal{G}(X)\|_H^2 \leq 2(\tau^2 K^2 + \|C\|^{\alpha-2}\epsilon_\tau^2)))$

$$\leq -\frac{3}{4}\int\left\|C^{\frac{\alpha}{2}}\nabla_{X_t}\log\frac{d\nu_t}{d\mu^{\mathbf{y}}}(X_t)\right\|_H^2 d\nu_t(X_t)$$

$$+ 64(t-k\gamma)^2\|C\|^\alpha L_{\mathcal{G}}^2\left(\int\left\|C^{\frac{\alpha}{2}}\nabla_{X_t}\log\frac{d\nu_t}{d\mu^{\mathbf{y}}}(X_t)\right\|_H^2 d\nu_t(X_t) + 2\text{Tr}(C^\alpha)L_{\Phi_0}\right.$$

$$\left. + 2(\tau^2 K^2 + \|C\|^{\alpha-2}\epsilon_\tau^2)\right) + 32\text{Tr}(C^\alpha)(t-k\gamma)L_{\mathcal{G}}^2 + 4(\tau^2 K^2 + \|C\|^{\alpha-2}\epsilon_\tau^2).$$

$$(23)$$

We can simplify (23) by letting $\gamma \leq \frac{1}{\sqrt{128\|C\|^\alpha L_{\mathcal{G}}}} \Rightarrow 64(t-k\gamma)^2\|C\|^\alpha L_{\mathcal{G}}^2 \leq 64\gamma^2\|C\|^\alpha L_{\mathcal{G}}^2 \leq \frac{1}{2}$.
Therefore, once $\gamma \leq \frac{1}{\sqrt{128\|C\|^\alpha L_{\mathcal{G}}}}$, we get

$$\frac{d}{dt}\text{KL}(\nu_t||\mu^{\mathbf{y}})$$

$$\leq -\frac{1}{4}\int\left\|C^{\frac{\alpha}{2}}\nabla_{X_t}\log\frac{d\nu_t}{d\mu^{\mathbf{y}}}(X_t)\right\|_H^2 d\nu_t(X_t)$$

$$+ 64(t-k\gamma)^2\|C\|^\alpha L_{\mathcal{G}}^2\left(2\text{Tr}(C^\alpha)L_{\Phi_0} + 2(\tau^2 K^2 + \|C\|^{\alpha-2}\epsilon_\tau^2)\right)$$

$$+ 32(t-k\gamma)\text{Tr}(C^\alpha)L_{\mathcal{G}}^2 + 4(\tau^2 K^2 + \|C\|^{\alpha-2}\epsilon_\tau^2).$$

$$(24)$$

By integrating (24) between $[k\gamma, (k+1)\gamma]$ we get

$$\text{KL}(\nu_{(k+1)\gamma}||\mu^{\mathbf{y}}) - \text{KL}(\nu_{k\gamma}||\mu^{\mathbf{y}})$$

$$\leq -\frac{1}{4}\int_{k\gamma}^{(k+1)\gamma}\left(\int\left\|C^{\frac{\alpha}{2}}\nabla_{X_t}\log\frac{d\nu_t}{d\mu^{\mathbf{y}}}(X_t)\right\|_H^2 d\nu_t(X_t)\right)dt$$

$$+ \frac{64}{3}\|C\|^\alpha L_{\mathcal{G}}^2\gamma^3\left(2\text{Tr}(C^\alpha)L_{\Phi_0} + 2(\tau^2 K^2 + \|C\|^{\alpha-2}\epsilon_\tau^2)\right) + 16\gamma^2\text{Tr}(C^\alpha)L_{\mathcal{G}}^2$$

$$+ 4\gamma(\tau^2 K^2 + \|C\|^{\alpha-2}\epsilon_\tau^2)$$

$$= -\frac{1}{4}\int_{k\gamma}^{(k+1)\gamma}\left(\int\left\|C^{\frac{\alpha}{2}}\nabla_{X_t}\log\frac{d\nu_t}{d\mu^{\mathbf{y}}}(X_t)\right\|_H^2 d\nu_t(X_t)\right)dt$$

$$+ \left(\frac{128}{3}\|C\|^\alpha L_{\Phi_0}\gamma + 16\right)\text{Tr}(C^\alpha)L_{\mathcal{G}}^2\gamma^2 + \left(\frac{128}{3}L_{\mathcal{G}}^2\|C\|^\alpha\gamma^2 + 4\right)\gamma(\tau^2 K^2 + \|C\|^{\alpha-2}\epsilon_\tau^2)$$

$$\leq -\frac{1}{4}\int_{k\gamma}^{(k+1)\gamma}\left(\int\left\|C^{\frac{\alpha}{2}}\nabla_{X_t}\log\frac{d\nu_t}{d\mu^{\mathbf{y}}}(X_t)\right\|_H^2 d\nu_t(X_t)\right)dt$$

$$+ \left(\frac{8\sqrt{2}}{3} + 16\right)\text{Tr}(C^\alpha)L_{\mathcal{G}}^2\gamma^2 + \left(\frac{1}{3} + 4\right)\gamma(\tau^2 K^2 + \|C\|^{\alpha-2}\epsilon_\tau^2),$$

where in the last inequality we invoked $\gamma \leq \frac{1}{\sqrt{128\|C\|^\alpha L_{\mathcal{G}}}}$ and the inequality $L_{\mathcal{G}} \geq \|C\|^{\alpha/2} L_{\Phi_0}$.

Now by averaging over $N > 0$ iterations and dropping the negative term, we can derive the result of Theorem 1:

$$\frac{1}{N\gamma} \int_0^{N\gamma} \left( \int \left\| C^{\frac{\alpha}{2}} \nabla_{X_t} \log \frac{d\nu_t}{d\mu^{\mathbf{y}}}(X_t) \right\|_H^2 d\nu_t(X_t) \right) dt$$

$$\leq \frac{4\mathrm{KL}(\nu_0\|\mu^{\mathbf{y}})}{N\gamma} + \left( \frac{32\sqrt{2}}{3} + 64 \right) \mathrm{Tr}(C^\alpha) L_{\mathcal{G}}^2 \gamma + \underbrace{\frac{52}{3} K^2}_{\substack{\text{Score Mismatch} \\ \text{Error}}} \tau^2 + \underbrace{\frac{52}{3} \|C\|^{\alpha-2}}_{\substack{\text{Score Approximation} \\ \text{Error}}} \epsilon_\tau^2.$$

$\square$

