# OpenReview forum: "Theoretical Convergence Analysis for Hilbert Space MCMC with Score-based Priors for Nonlinear Bayesian Inverse Problems"
_ICLR.cc/2025/Conference — ICLR 2025 Conference Withdrawn Submission_

### Official Review · Reviewer_BsDj · 2024-10-29

**Soundness:** 3
**Presentation:** 2
**Contribution:** 3
**Rating:** 5
**Confidence:** 3

**Summary:**

This paper studies the nonlinear Bayesian inverse problems using Hilbert space MCMC with score-based priors. It provide theoretical guarantees for the Hilbert space Langevin-type MCMC algorithm  by conducting a non-asymptotic stationary convergence analysis.

**Strengths:**

1. It present a theoretical analysis of the Langevin-type MCMC algorithm  with score-based generative priors in Hilbert space, extending  previous results from finite dimension to an infinite-dimensional  setting.

2. The bound obtained reveal an interplay between the preconditioner and score approximation error,  which is novel.

**Weaknesses:**

1. The technical novelty is not clearly stated. As the work extends  Sun. et al (2024) to infinite dimensions, it is better to discuss the extra technical challenge addressed in the paper and the novel aspects of the proof.

2. The expression for the bound in Theorem 1 is complicated and it is hard to see the theoretical interpretation of the result. It is better to provide an intuitive explanation for each term. How does the choice of $\alpha$ affect the result? What will happen if $\alpha$ is near zero or $\alpha$ is extremely large? How do $\tau,\gamma$ influence the bound and is there an optimal choice for them?

3. The Assumption 3, which assumes the score network is  Lipschitz continuous with a Lipschitz constant $L_{\tau}$, this seems problematic as for deep neural networks the  Lipschitz constant can increase exponentially with depth.  Why the assumption is necessary?

**Questions:**

Please see the weakness part.

---

### Official Review · Reviewer_guFZ · 2024-11-03

**Soundness:** 2
**Presentation:** 3
**Contribution:** 2
**Rating:** 3
**Confidence:** 3

**Summary:**

This study utilizes a newly established framework for score-based generative models in Hilbert spaces to acquire an infinite-dimensional score. This score is subsequently employed as a prior in a function-space Langevin-type MCMC algorithm, offering theoretical assurances for convergence within nonlinear Bayesian inverse problems. The research demonstrates that managing the approximation error of the score is crucial not only for guaranteeing convergence but also for indicating the necessity of adjusting the conventional score-based Langevin MCMC approach via the adoption of a suitable preconditioner.

**Strengths:**

The problem of theoretical foundations of hilbert space mcmc with score-based priors is meaningful.
The presentation is clear.

**Weaknesses:**

1. The paper is fully theoretical. It remains to see if it has practical guidance to real hilbert space SGM algorithms. It is anticipated to include at least some numerical illustrations to explain your "theorems", otherwise I don't think it lives up to a top conference paper.

**Questions:**

None.

---

### Official Review · Reviewer_AXPz · 2024-11-03

**Soundness:** 3
**Presentation:** 3
**Contribution:** 2
**Rating:** 3
**Confidence:** 3

**Summary:**

The paper presents an algorithm which might be viewed as a natural infinite-dimensional analogue of PMC-RED. As noted, there are specific considerations when dealing with functional spaces/infinite dimensional settings which do not occur in finite dimensional contexts and so dedicated work is necessary to justify the use of algorithms in these settings.

It provides a theoretical analysis of this infinite-dimensional algorithm which takes the form of Theorem 1, providing explicit bounds on the error of such a scheme when using score-based priors under a collection of assumptions. It is established that preconditioning is required (experience with other MCMC algorithms adapted to infinite dimensional spaces suggest that this is not a surprising finding) and that it's necessary to control the error in the approximation of the score in order to obtain the established convergence.

As is usual when dealing with infinite dimensional algorithms, this necessitates the development of non-trivial error bounds which do not depend upon the dimension of the space.

**Strengths:**

The paper provides a rigorous convergence analysis, and finite sample quantitative error bounds (with interpretable components) for a Langevin-type algorithm applied in infinite dimensional settings in which score-based priors are utilised. Providing theoretical guarantees for such algorithms seems to me to be an important contribution.

Rigorous analysis in functional settings does seem to be missing from this part of the literature at present and a step towards providing it is valuable.

**Weaknesses:**

I was surprised not to see any discussion of how to implement (6) in practice: in the context of "proper" Hilbert spaces, i.e. excluding finite dimensional inner product spaces, it does not seem as though a direct implementation would be possible without further approximations unless I've misunderstood something (certainly possible). The concern that I have about the relevance of the paper largely comes down to this: a formal analysis of (6) only constitutes half of the story without supplementing it with some analysis of the impact of relating (6) to an implementable algorithm; if I have misunderstood something here, or the authors are able to reassure me that this analysis really does tell us something about an implementable algorithm then I would be substantially more positive in my assessment.

The structure of the proof doesn't make it easy to gain much insight into the arguments without a line by line reading of the argument; although a very brief outline of the structure of the proof of the main theorem is presented in the main manuscript, the argument given in the supplementary material in detail is rather lacking in linking text and consists primarily of quite a number of pages of algebra. As a major goal of the manuscript appears to be to encourage the community to pay more attention to the requirements of infinite dimensional settings it would have been nice if it was easier for the casual reader to get more of a sense of what matters here and how the difficulties are addressed. (As one example, 791-793: "Before going through the proof of Theorem 1, we will need two lemmas. Similar results have been
proved in (Sun et al., 2024; Balasubramanian et al., 2022; Vempala and Wibisono, 2019) for the
finite-dimensional setting.". Why the lemmas are needed; what they establish and how would be useful things to communicate to the reader at this point.)

There are no examples. I don't necessarily believe that theoretical papers require numerical validation; the existence of a theorem with a rigorous proof is validation enough. However, examples of settings of interest in which the proposed algorithm could in principle be used (and, ideally, the assumptions verified) would have been helpful to most readers and some illustration that the proposed approach has (or does not have) superior performance than methods which have been proposed without rigorous validation would also strengthen the paper in my view.

A few minor typos etc. that don't have any significant impact:
line 219: "PCM-RED"
243 "a distribution with respect to the Lebesgue measure" distribution -> density?

**Questions:**

How do you propose to implement the proposed algorithm for real problems? Presumably some kind of basis expansion or other finite dimensional representation needs to be combined with (6) in order to admit actual sampling? At some point a finite dimensional representation must be available to allow computational implementation.

Are you able to demonstrate that whatever steps need to be taken to turn (6) from a mathematical algorithm to an implementable one do not harm the convergence properties which are established?

232: "one can interpret (3) as a parallel MCMC algorithm of RED for posterior sampling" do you mean a parallel MCMC algorithm (in the computational sense), as this phrasing seems to suggest, or something more like as an analogue?

 248: "Here and throughout the paper, we assume that the prior µ0 that we want to learn from data to perform Bayesian inference in (1) is the Gaussian measure µ0 = N (0, Cµ0 ), though our analysis can be easily generalized to other classes of priors, e.g. priors given as a density with respect to a Gaussian" raises for me a number of questions:
a) in Bayesian inference the prior isn't something to be learned from data; perhaps you have in mind something akin to empirical Bayes?
b) if you need to use score based techniques to learn some reasonable prior structure then how reasonable is it to suppose that you can be confident about its form to the degree necessary to apply these results?

---

### Official Review · Reviewer_GQpm · 2024-11-04

**Soundness:** 3
**Presentation:** 3
**Contribution:** 2
**Rating:** 3
**Confidence:** 2

**Summary:**

This paper proposes a variant analysis of the score-based denoising diffusion models but for an infinite dimensional Hilbert space. An analogue of the standard convergence results are presented, where various generalizations of the standard assumptions are used (for instance, the score error is now with respect to the norm of said space).

**Strengths:**

The paper contains a rigorous generalization of the theory and assumptions for score-based generative models in the case where the distribution is over an infinite dimensional Hilbert space.

The result seems to be easily extended to algorithmic modifications, such as different discretizations, etc.

Some leg-work is needed to make the result rigorous in infinite dimensions due to unique technical issues in that setting. It seems that the bulk of the proof is dedicated to this.

**Weaknesses:**

The results do not seem particularly surprising, and appear to simply generalize what is known from the finite dimensional case.

I think a bit more effort has to be presented to justify why this is an important regime of interest. In particular, why is such a result useful when the parametrizations chosen in practice need to be finite dimensional? In particular, it seems sufficient to project the result onto a sufficiently large finite dimensional subspace carrying most of the mass of the prior/target. Instead, perhaps the authors should focus on why such infinite-dimensional results can imply dimension-free results in a finite dimensional setting.

**Questions:**

What notion of derivative in being used in this Hilbert space (in the definition of the Fisher information and elsewhere)? The Fréchet derivative?

Can the authors clarify the role of Assumption 4 relative to prior work in the literature? Is this a reasonable factorization or does it seem to impose a strong constraint upon the target distribution?

In line 069, do the authors mean to learn the score of the posterior (rather than prior)?

In Line 121, what do the authors mean by "keeping an explicit likelihood"?

In Line 223, can you clarify what $g$ is the negative log-likelihood of?

In Line 224, can you clarify the sense in which $\nabla \log p_\sigma$ converges to $\nabla \log p$?

---

### Note · Authors · 2024-11-14

I have read and agree with the venue's withdrawal policy on behalf of myself and my co-authors.